

# 1 Geogenic organic carbon in terrestrial sediments and its
# 2 contribution to total soil carbon

Fabian Kalks[a]*, Gabriel Noren[b], Carsten Mueller[cd], Mirjam Helfrich[a], Janet Rethemeyer[b], Axel Don[a]
[a] Thünen Institute of Climate-Smart Agriculture, Bundesallee 65, 38116 Braunschweig, Germany
[b] Institute of Geology and Mineralogy, University of Cologne, Institute of Geology and Mineralogy, Zülpicher
Str. 49b, 50674 Cologne, Germany
[c] Technical University of Munich, Chair of Soil Science, Emil-Ramann Strasse 2, 85354 Freising-Weihenstephan
[d] University of Copenhagen, Department of Geosciences and Natural Resource Management, Øster Voldgade 10,
DK-1350 Copenhagen K, Denmark
*Corresponding author: fabian.kalks@thuenen.de + 49 531 596 2719

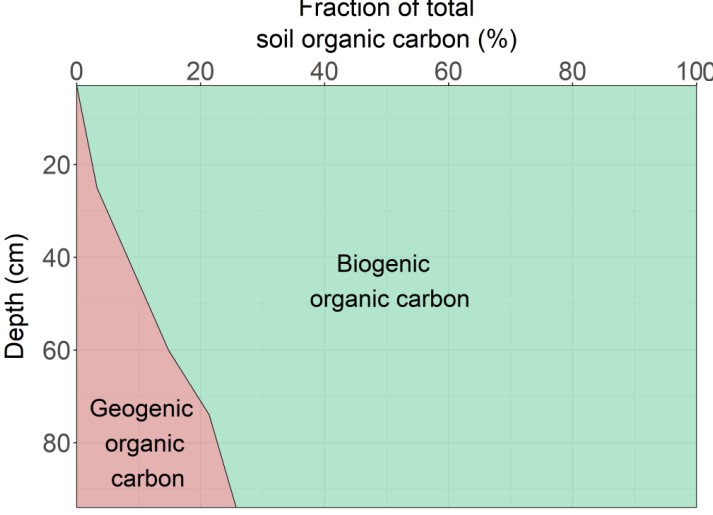

16  **Graphical abstract:** The median GOC contribution (based on all three sediments) to soil OC with depth based on [14]C
17  measurements.



## Abstract

Geogenic organic carbon (GOC) from sedimentary rocks is an overlooked fraction in soils that has not been quantified yet, influencing the composition, age and stability of total organic carbon (OC) in soils. In this context GOC is referred to as the OC in bedrocks deposited during sedimentation. However, the contribution of GOC to total soil OC varies with the type of bedrock. So far studies investigating the contribution of GOC derived from different terrestrial sedimentary rocks to soil OC contents are missing.

In order to fill this gap, we analysed 10 m long sediment cores at three sites recovered from Pleistocene Loess, Miocene Sand and Triassic Red Sandstone and calculated the amount of GOC based on $^{14}C$ measurements. $^{14}C$ ages of bulk sedimentary OC revealed that OC represents a mixture of biogenic and geogenic components. Biogenic refers to OC that entered the sediments recently from plant sources. All sediments contain considerable amounts of GOC (median amounts of 0.10 g kg$^{-1}$ at the Miocene Sand, 0.27 g kg$^{-1}$ at the Pleistocene Loess and 0.17 at Red Sandstone) in comparison to subsoil OC contents (between 0.53-15.21 g kg$^{-1}$). Long-term incubation experiments revealed that this GOC seemed to be comparatively stable against biodegradation. Its possible contribution to subsoil OC stocks (0.3-1.5 m depth) is ~ 2.5 % in soil developed in the Miocene Sand, ~ 8 % in the Loess soil and ~ 12 % at the Red Sandstone site. Thus GOC having no detectable $^{14}C$ contents influences $^{14}C$ ages of subsoil OC and thus may partly explain the strong $^{14}C$ ages increase observed in many subsoils. This is particularly important in soils on terrestrial sediments with comparatively low amounts of OC, where GOC can considerably contribute to total OC stocks

**Keywords** Geogenic organic carbon; sedimentary organic carbon; $^{14}C$; terrestrial sediments; incubation experiment



## 1. Introduction

On a global average soils store more than 50 % of OC in the subsoil below 30 cm depth (Batjes, 2014). This carbon is considered as a highly stabilised carbon pool due to its high apparent [14]C ages (Schrumpf et al., 2013, Mathieu et al., 2015). This, however, may also be explained by an contribution from geogenic organic carbon (GOC), which is defined here as OC that is deposited during sedimentation and rock formation, and may increasingly influence subsoil OC with increasing depth (Schrumpf et al., 2013, Graz et al., 2010, Kögel-Knabner et al., 2008, Trumbore, 2009). GOC in most cases is devoid of [14]C and thus may lead to an overestimation of ancient OC sources although a number of studies showed the importance of root derived, young OC inputs to subsoils (Angst et al., 2016, Crow et al., 2009). The contribution of GOC to soils has been investigated in reclaimed mine soils where Vindušková et al. (2015) found contributions from GOC between 26 and 99 % to total soil OC. Furthermore especially OC rich sediments with contents of 2-7 g kg$^{-1}$ (Hemingway et al., 2018) or 28-105 g kg$^{-1}$ (Frouz et al., 2011) have been investigated. The impact of GOC on soils derived from sediments or sedimentary rocks with lower OC contents, however, has not been investigated so far. Considering the fact that approximately 65 % of the continental earth's surface is covered with sediments and sedimentary rocks (Amiotte Suchet et al., 2003) a potentially large fraction of soils could contain GOC that contributes to soil OC stocks, even though a lot of them might be derived from recent sedimentation processes. So far there is not much literature about sediments with only low amounts of OC. There are estimations that assume sandstones to be GOC free (van der Voort et al., 2018) or, in contrast, a storage model that assumes generally high GOC amounts of 2.4 g kg$^{-1}$ for all sandy deposits (Copard et al., 2007). Therefore, more information about the amounts of OC in sediments is needed.

To estimate the possible contribution of GOC to subsoil OC stocks, it is further necessary to know about the amount of OC in sediments that comes from sedimentation (GOC) and to distinguish it from OC that is derived from current vegetation (biogenic OC). There are many soil- and substrate-specific factors that might influence the OC contribution from current vegetation to sedimentary OC like potential rooting depth or hydraulic conductivity. So far no method could be established that would allow a direct quantification of GOC in different soils or sediments, apart from promising methods to quantify the graphitic part of GOC in soils (Zethof et al., 2019). The only reliable approach to distinguish both sources is the use of [14]C. Because deposition of sediments mostly took place > 50,000 yrs. BP, they do not contain any [14]C, which has a mean half life time of 5,730 yrs. (Libby, 1952). In addition, δ$^{13}$C values of OC in the sediments allow to distinguish carbonaceous from organic sources. Thus, using both carbon isotopes can reveal if the OC is a mixture of GOC and OC from the vegetation that is younger than 50,000 yrs. A quantification of the geogenic part of OC in the sediments is only possible if the average [14]C age of biogenic OC is known or can be estimated.



One important question regarding a possible contribution from GOC in soils is, if this GOC will be
mineralised when it becomes part of the soil. Due to the fact that GOC resists degradation since it has
been deposited, it can be assumed that it already exhibits a strong inherent recalcitrance. Nevertheless,
this could also be due to a physical protection that prevented microbial accessibility. However, when it
becomes part of the subsoil during progressing soil development, the infiltration of water, oxygen,
fresh nutrients and microorganisms might cause the degradation of this OC pool. The direct microbial
coal degradation has already been observed via incubation experiments in mine soils (Waschkies and
Huttl, 1999, Rumpel and Kögel-Knabner, 2002) or in shale bedrocks directly exposed to the surface
(Soulet et al., 2017). If GOC is degradable in OC-poor sediments or sedimentary rocks has not been
investigated so far but might be different since the amount of OC can also drive microbial respiration
(Colman and Schimel, 2013).
To the best of our knowledge there is only one study by van der Voort et al. (2018) investigating the
amount of GOC in soils and estimating this to make up about 80 % of soil OC in a moraine derived
soil. This reveals that GOC might considerably contribute to soil OC. But beside the study from van
der Voort et al. (2018) on a very specific sediment, further direct calculations of the amount of GOC in
soils are missing.
Our aim was to quantify GOC in different terrestrial sediments and a sedimentary rock and investigate
its stability in incubation experiments to make assumptions about its possible contribution to soil OC
stocks in soil profiles at the same site. Our main research questions were i) what is the relation
between sedimentary and subsoil OC contents? ii) is OC in sediments $^{14}$C free and how much is really
geogenic? iii) will sedimentary GOC be degraded? and iv) how much does GOC contribute to soil
OC?



## 2. Material and methods

### 2.1 Site description

Three sites were selected with different sedimentary bedrocks derived from a single geologic substrate, that can be found close to the surface and that is homogeneous down to 10 m depth. The sites represented one sedimentary rock and two soft sediments. The sedimentary rock was a sandstone (Solling Formation, Triassic) under European beech forest (*Fagus sylvatica*) 11.5 km north-east of Göttingen (51°35.012' N; 10°3.960' O) in the following referred to as "Red Sandstone". The soil was classified as a Cambisol according to World Reference Base for Soil Resources (WRB, 2006). The sediments were loessic deposits (Weichselian Glacial) under an agricultural field, 30 km north of Göttingen (51°48.101 N; 9°58.002' O) referred as "Loess" and terrestrial sandy deposits from the Miocene (Neogene formerly named Tertiary) in a European beech forest 13 km south-west of Göttingen (51°28.673 N; 9°45.323' O) referred to as "Miocene Sand". The associated soils are classified as a Luvisol and a Cambisol respectively. Mean annual air temperature and precipitation were 9.2 °C and 647 mm (1981-2010) at the nearby weather station including all three sites.

### 2.2 Sampling and sample preparation

Two 10 m long sediment cores with a diameter of 15 cm were drilled at each site in April 2017. For the soft sediments, drilling was conducted as percussion drilling and for the hard sediments as cable core drilling with water as flushing solution. We subdivided the sampled cores into 1 m increments. The replicates per site were drilled in a distance of approximately 10 m.

One sample from each depth increment (1-2 m, 2-3 m, 3-4 m , 4-5 m, 5-6 m, 6-7 m, 7-8 m, 8-9 m, 9-10 m) for chemical analysis was oven dried at 60°C and sieved to pass 2 mm. The Red Sandstone samples were crushed with a hammer before drying and sieving was conducted. To avoid a possible contamination the outer 5 cm of the drilling cores were removed. Additionally, 1 m deep soil profiles were dug and soil samples were taken from the different classified layers to obtain corresponding soil parameters. Samples were oven dried at 60°C and stored and sieved to pass 2 mm. For the determination of OC and $^{13}$C samples were ground in a planetary ball mill. For $^{14}$C analysis, subsamples were decarbonized with 1M HCl and heating for 1 h at 80 °C followed by 10 h at room temperature.

### 2.3 Chemical analysis and calculations

Three aliquots of each sieved sample was analysed by dry combustion for total C and total N content (TruMac CN LECO, St. Joseph, MI, USA). Samples with a pH value of > 6 were analysed for carbonates after ignition of the sample at 450 °C for 16 h in a muffle kiln. The OC concentration was calculated by subtracting carbonaceous C from total C and expressed as g OC kg$^{-1}$ dry matter. Homogenised samples were further analysed for $\delta^{13}$C values after removing carbonates in an isotope





ratio mass spectrometer (Delta Plus, Thermo Fisher, Waltham, MA, USA) coupled to an elemental
analyser (FLASH EA 1122 NA 1500; Wigan, United Kingdom). Resulting $\delta^{13}C$ values (‰) were
expressed relative to the international standard of Vienna Pee Dee Belemnite. The bulk densities for
the soil samples were obtained with 250 cm³ sampling rings from each layer of the soil profile. For the
sedimentary Loess and Miocene Sand samples, the bulk density of the deepest respective soil sample
was used. The bulk densities and the densities without pore space of the intact Red Sandstone cores
were determined on four subsamples (from 1.6, 3.6, 7 and 9 m depth) with a Dryflow-pycnometer
(GeoPyc 1360) and a gas pycnometer (AccuPyc 1330) respectively. Bulk densities for the missing
depth increments were linearly interpolated. . For radiocarbon ($^{14}C$) analysis, the sediment samples
were first treated with acid to remove inorganic C and where then transferred into pre-combusted
quartz ampoules together with copper oxid and silver wool. The ampoules were evacuated, flame
sealed, then combusted at 900°C, and the $CO_2$ evolved was purified on a vacuum ring (Rethemeyer et
al., 2019). $^{14}C$ contents were measured with the MICADAS accelerator mass spectrometry (AMS)
system at the ETH Zürich, Switzerland. If possible one sedimentary sample per depth increment and
site and one sample per soil layer was analysed. Due to the very low OC contents in some sediment
samples, $^{14}C$ contents could only be determined for three samples from the Miocene Sand (from 1.9,
4.9 and 7.9 m depth) and four from the Red Sandstone (1.9, 4.9, 7.9 and 9.9 m depth). For the Loess,
$^{14}C$ of bulk OC was measured in all depth intervals (1.9-9.9 m).
Total OC stocks (Mg ha$^{-1}$) were calculated according to Eq. 1:

$$OC\ stock = OC \cdot BD \cdot (1 - stone\ content) \cdot depth \cdot 0.1 \qquad \text{Eq. 1}$$

where *OC* is the weight based OC content, either in the fine soil <2-mm fraction of the soil profiles, or
in the sediments (g kg$^{-1}$), *BD* is the bulk density of the fine soil (g cm$^{-3}$), stone content is the volume
based proportion of stones (cm³ cm$^{-3}$) and depth is the thickness of the depth increment (cm). Due to
the fact that we did not determined the transition from soil to sediment exactly we set it to 1.5 m depth
for all sites. This represents a common boarder for the transition from soil to sediment according to
Richter and Markewitz (1995). This was done to be able to compare OC stocks and contributions from
GOC later on. We further subdivided the sediments into an upper and a lower part at 4 m depth.
In a second step we calculated the amount of GOC and biogenic OC in the sediments considering
GOC as one carbon pool free of $^{14}C$. For the sediments we calculated the proportion of biogenic OC
($f_{biogenic}$) on the total amount of OC with a two pool model (Eq. 2) used by Cerri et al. (1985):

$$f_{biogenic}\ (\%) = \frac{F_{biogenic\ OC} - F_{GOC}}{F_{sample} - F_{GOC}} \cdot 100 \qquad \text{Eq. 2}$$

where *F* represents the $^{14}C$ content in fraction modern carbon ($F^{14}C$) from a source compared to the
$^{14}C$ content of an oxalic standard (Torn et al., 2009, Stuiver and Polach, 1977). Sources were the GOC



fraction ($F_{GOC}$), the sample ($F_{sample}$) and the biogenic OC fraction ($F_{biogenic\ OC}$). Since the [14]C content of
the GOC fraction can be set to zero, this equation can be simplified to:

$$f_{biogenic}\ (\%) = \frac{F_{biogenic\ OC}}{F_{sample}} \cdot 100 \qquad\qquad \text{Eq. 3}$$

For the biogenic OC in the sediments we assumed an average [14]C age ranging from 1,000-4,000 yrs.
BP. We assumed this range based on published [14]C results of dissolved OC reaching greater depths
(Schiff et al., 1997, Artinger et al., 1996). The [14]C contents in the sediment from 2 to 4 m depth of the
loess led to ages < 3,000 yrs. BP and were therefore even younger than in 74 cm depth (4,413 yrs.
BP). Thus, they were treated like the soil part for the calculation of a GOC fraction in the following.
Respective times were converted into [14]C contents ($F_{biogenic\ OC}$) according to Torn et al. (2009):

$$F_{biogenic\ OC} = e^{\left(\frac{t}{-8033}\right)} \qquad\qquad \text{Eq. 4}$$

where $t$ represents the [14]C age (1,000 or 4,000 yrs. BP respectively) and 8033 yrs. the mean life of
radiocarbon. The proportion of GOC in the sediments ($f_{GOC}$) consequently is the remaining portion
(Eq. 5).

$$f_{GOC} = 100\ \% - f_{biogenic} \qquad\qquad \text{Eq. 5}$$

For the depth increments without measured [14]C ages (Tab. S1) we linearly interpolated the calculated
amounts with measured [14]C ages from over- and underlying samples. This was done by assuming a
depth dependent correlation and using the adjacent values. To calculate the amount of GOC in the soil
profiles we first calculated the weight-based amount of GOC in the sediments by multiplying its
fraction ($f_{GOC}$) with the respective OC content (in g OC per kg[-1] dry mass). We then took the median
amount of GOC (g GOC per kg[-1] dry mass) of these sedimentary values and calculated its proportion
on the  soil OC content (g OC per kg[-1] fine soil) in the soil profile. This was done for the proportion of
GOC in the sediments calculated with a 1,000 and a 4,000 year old biogenic OC fraction ($F_{biogenic\ OC}$ in
Eq. 3) to obtain a range of GOC contributions. We assumed that the GOC fraction resisted degradation
during soil formation. Therefore, this proportion represents the highest possible amount of GOC that
may contribute to soil OC stocks. Under this assumption we were also able to define the influence of
GOC in the soil profile on the resulting [14]C ages. Since the calculated [14]C ages represent a mixture of
the [14]C content from the GOC and the biogenic fraction (Eq. 5), the GOC fraction has the same
influence on the soil [14]C age as on bulk OC (according to Eq. 3). Reducing the age by this fraction
would therefore represent an "unbiased" age of soil derived OC.
*2.4 Incubation experiment*
To assess the potential stability of OC in the sediments against microbial decay, two laboratory
incubation experiments were conducted at 20°C for 50 and 533 days respectively. This was done to





reveal the potential degradation of OC from the sediments under optimal conditions. The first, 50 days
lasting experiment was conducted with intact Red Sandstone core samples, while the second
experiment was performed after crushing the Red Sandstone to sizes < 2 mm. This was done to
simulate the process of weathering when the intact sediment or sedimentary rock becomes part of the
(sub)soil. The Loess and Miocene Sand samples were flushed with fresh air between both incubations.
For the incubations, four subsamples with 1,340-6,890 g per sample from different depth intervals
were used. The sample material was stored at room temperature until the incubation experiment
started. We took four samples from each sediment and from each of four depth ranges for the Miocene
Sand (1.2-2.8, 3.2-4.8, 6.1-7.8 and 8.2-9.9 m), the Red Sandstone (2-3, 4-5.8, 7-8 and 8.4-9.8 m) and
the Loess (1.4-2, 2.7-3, 4.8-6 and 9.1-10 m). Additionally four blank samples with no material were
installed. A water content corresponding to 40 % of the water holding capacity based on the pouring
density was adjusted. Based on preliminary tests and its calculated bulk density and porosity, the intact
Red Sandstone samples were kept in a barrel with pure water for 14 hours to reach a water content of
nearly 40 %. Samples were placed in polycarbonate vessels with a volume of 7069 cm³ and closed air-
tight. The lids contained two tube connectors so that the samples could be flushed with ambient air.
After flushing, samples were set to a starting pressure of about 1,300 mbar and kept closed until the
end of the incubation. Nine gas samples were taken in evacuated glass vials (20 mL), 0, 3, 7, 13, 21,
30, 59, 63 and 533 days after the incubation started. Samples were analysed for $CO_2$ concentrations by
gas chromatography (Agilent 7890A, GC, Agilent Technologies, Santa Clara, USA) to account for the
amount of accumulated $CO_2$. Three additional gas samples were taken 0, 30 and 63 days after the
incubation started and analysed with an isotope ratio mass spectrometer (Delta Plus XP, Thermo
Fisher Scientific, Bremen, Germany) to account for the development of $\delta^{13}C$ of $CO_2$ during the
respiration. Corresponding pressure was measured at each sampling date. When the over-pressure of a
vessel was lost due to leakages, it was removed from the sampling because a contamination with
ambient air could not be excluded. This happened for one third of all samples.
The amount of respired $CO_2$-C (mg $CO_2$-C d$^{-1}$) was calculated with Eq 6.

$$CO_2 - C = \frac{0.1 \cdot p \cdot x_i \cdot M \cdot V}{R \cdot T \cdot t} \qquad \text{Eq. 6}$$

where $p$ is the pressure (mbar), $x_i$ is the difference of the $CO_2$ concentration between the samplings
(ppm), $M$ is the molar mass of C (g mol$^{-1}$), $V$ the air volume of the sample (m³), $R$ is the molar gas
constant (J kmol$^{-1}$ K$^{-1}$), $T$ is the incubation temperature (K) and $t$ is the elapsed time (d) between the
samplings. Based on the $\delta^{13}C$ values of $CO_2$ the proportion of $CO_2$ derived from carbonates was
subtracted according to Bertrand et al. (2007) if necessary. This respiration rate was related to the OC
content of the samples (called "OC-normalised respiration") by dividing it by the total amount of OC
in g in the sample.





Assuming that OC in the sediments represents a mixture of a stable and a labile pool with different
extents of degradability, we fitted a double exponential model to this OC-normalised respiration with
different rates according to Qualls and Haines (1992) :

$$mineralised\ OC\ (\%) = (100 - a) \cdot \left(1 - e^{-k1 \cdot t}\right) + a \cdot \left(1 - e^{-k2 \cdot t}\right) \qquad \text{Eq. 7}$$

where $a$ represents the proportion of the stable OC pool (%), $k1$ and $k2$ the associated mineralisation
rate constants of the labile and stable OC (year$^{-1}$) and $t$ the elapsed time (years). We also calculated the
mean residence time (years) of the labile and the stable OC pool ($1/k1$ and $1/k2$). If it was not possible
to fit a double exponential model we used a simple linear regression (*mineralised OC (%) = k · t*) to
obtain a mineralisation rate. This was the case for the Red Sandstone samples, showing a linear
mineralisation rate during the incubation experiment.
*2.5 Statistics*
Statistical analyses were conducted using the statistical environment R (R Core Team (2018) including
the base function "*nls*" (nonlinear least-squares estimates) to fit non-linear models, "*lm*" to fit linear
models and the package *ggplot2* (Wickham, 2016) for graphical presentation. The non-linear models
were used to model the amount of mineralised OC during the incubation experiment using Eq. 7.
Models were tested for deviations from homoscedasticity, normality of residuals and absence of
collinearity. The tests revealed heteroscedasticity of the residuals. This can be explained by an
increasing standard deviation with time. We are further aware of not having normally distributed
residuals since the dependent variable, representing the proportion of OC being mineralised, only
allows values between 0 and 1. Keeping this in mind the results from the double exponential model
have to be treated as an indicator for the differences between the samples and a scale for the
mineralisation of a stable and a labile OC pool. We therefore also omitted calculations of standard
errors and significance of the parameters since this would not lead to reasonable results with the used
models.





## 3. Results

*3.1 Relation between sedimentary and subsoil organic carbon*

In all analysed sediments OC contents were measured above the detection limit. The amount of OC in the sediments from 1 to 10 m depth was comparatively low in the Miocene Sand and Red Sandstone (0.04-0.71 g kg$^{-1}$ and 0.01-0.53 g kg$^{-1}$, respectively). Considerably higher OC contents of 0.21-9.71 g C kg$^{-1}$ were found in the Loess (Fig. 1 a). The median OC content of the sediments was in a comparable range like those in the respective deepest subsoil horizon. This deepest horizon was a Cv horizon in 94 cm depth for the Loess, 100 cm depth for the Miocene Sand and 74 cm depth for the Red Sandstone. In detail, the median OC content in the sediments, compared to the respective Cv horizons, corresponded to 27 % for the Loess, 29 % for the Miocene Sand and 39 % for the Red Sandstone. The Loess OC contents were highly variable, highlighting the changing sedimentary conditions during the past glacial and interglacial periods (Jordan and Schwartau, 1993). In 4-5 m depth, OC contents of the Loess were even higher (9.7 g kg$^{-1}$) than in the subsoil (3 g kg$^{-1}$). In the Miocene Sand and the Red Sandstone no clear depth gradient of OC was found in 2-10 m depth (Fig. 1 a). Even though OC content in the sediments are low, OC stocks can be considerable large. A comparison of OC stocks in topsoils (0-0.3 m), subsoils (0.3-1.5 m) and the sediments down to 10 m depth revealed quite high OC stocks in the sediments. For the Loess, OC in the sediment contributed up to 71 % of the total OC amount while it was 51 % for the Red Sandstone and 21 % for the Miocene Sand (Table 1).

The distribution of the $\delta^{13}$C values of OC in the soil and sediment profiles showed an increase of $\delta^{13}$C with increasing depth in the soil down to 1 m depth (Fig. 1 b). Contrastingly, the $\delta^{13}$C values of OC in the sediments showed no clear trend with increasing depth but they all were in the range of C$_3$ plant material. A value above -25 ‰ for the Red Sandstone in 4 m depth can be explained by corresponding high values of inorganic carbon in this depth. It can be assumed that decarbonisation of this sample was not completely successful. Unexpectedly high amounts of inorganic carbon were found in parts of the Red Sandstone, indicating the presence of calcareous deposits in this terrestrial material (Fig. 1 c). The Loess also includes some distinct calcareous layers in 5 and 10 m depth, while there was no inorganic carbon present in the Miocene Sand.





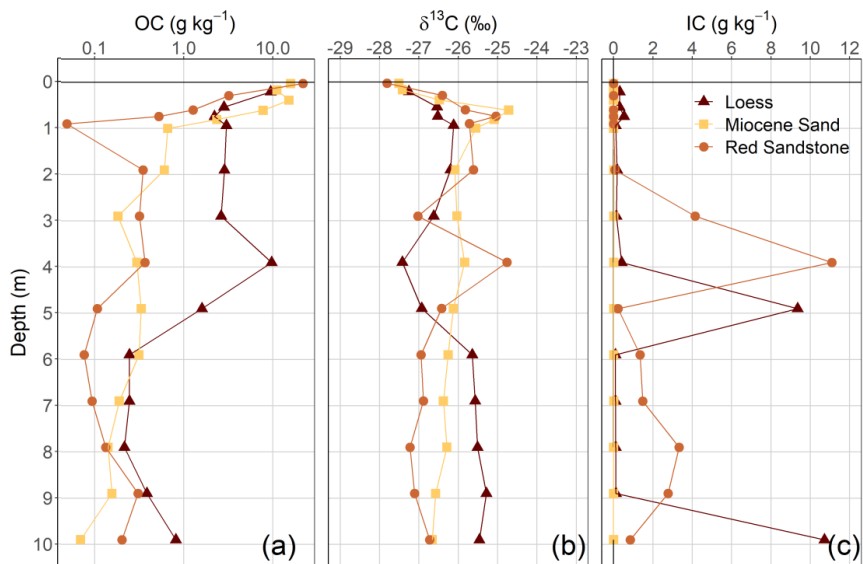

286

**Fig. 1:** Depth distribution of different bulk properties of the soil profiles and the deep drilling cores. Presented

parameters include the log scale organic carbon (OC) content (a), $\delta^{13}C$ values of the organic carbon (b) and the

inorganic carbon (IC) content (c) related to the amount of fine soil or dry mass respectively.

**Table 1:** OC stocks and proportions for the three sites down to 10 m depth. Proportions of biogenic and

geogenic OC were calculated based on [14]C results and assumptions described in the material and methods

section. Represented ranges are calculated based on the assumption of a 1,000 or 4,000 year old biogenic OC

fraction reaching the sediments.

| Substrate | Layer | Depth (m) | TOC (Mg·ha[-1]) | (%)[a] | OC stocks (Mg ha[-1]) geogenic 4,000 yrs. | geogenic 1,000 yrs. | biogenic 1,000 yrs. | biogenic 4,000 yrs. | Proportion of OC (%)[b] geogenic 4,000 yrs. | geogenic 1,000 yrs. | biogenic 1,000 yrs. | biogenic 4,000 yrs. |
|---|---|---|---|---|---|---|---|---|---|---|---|---|
| Loess | Topsoil | 0.0 - 0.3 | 40.6 | 11 | 1.1 - 1.2 | | 40 - 39 | | 3 - 3 | | 97 - 97 | |
| | Subsoil | 0.3 - 1.5 | 66.0 | 17 | 5.1 - 5.3 | | 61 - 61 | | 8 - 8 | | 92 - 92 | |
| | Upper Sediment | 1.5 - 4.0 | 218.6 | 57 | 10.6 - 11.0 | | 208 - 208 | | 5 - 5 | | 95 - 95 | |
| | Lower Sediment | 4.0 - 10.0 | 55.1 | 15 | 42.9 - 46.7 | | 8 - 12 | | 78 - 85 | | 15 - 22 | |
| Red Sandstone | Topsoil | 0.0 - 0.3 | 22.2 | 30 | 0.6 - 0.7 | | 22 - 22 | | 3 - 3 | | 97 - 97 | |
| | Subsoil | 0.3 - 1.5 | 13.7 | 19 | 1.6 - 1.7 | | 12 - 12 | | 12 - 12 | | 88 - 88 | |
| | Upper Sediment | 1.5 - 4.0 | 18.7 | 25 | 13.1 - 14.9 | | 4 - 6 | | 70 - 80 | | 20 - 30 | |
| | Lower Sediment | 4.0 - 10.0 | 19.3 | 26 | 14.1 - 15.7 | | 4 - 5 | | 73 - 82 | | 18 - 27 | |
| Miocene Sand | Topsoil | 0.0 - 0.3 | 39.1 | 31 | 0.3 - 0.3 | | 39 - 39 | | 1 - 1 | | 99 - 99 | |
| | Subsoil | 0.3 - 1.5 | 60.8 | 48 | 1.3 - 1.6 | | 59 - 60 | | 2 - 3 | | 97 - 98 | |
| | Upper Sediment | 1.5 - 4.0 | 10.6 | 8 | 3.7 - 5.8 | | 5 - 7 | | 34 - 55 | | 45 - 66 | |
| | Lower Sediment | 4.0 - 10.0 | 16.3 | 13 | 8.3 - 10.8 | | 6 - 8 | | 51 - 66 | | 34 - 49 | |

294

[a] % of total OC stock from 0-10 m

[b] % of OC stock in the respective depth increment





*3.2 Ages of organic carbon in soils and sediments and contributions from geogenic organic carbon*

The ages of OC in the Loess soil profiles revealed a modern carbon like signature (0 yrs. BP) in 0.3 m depth with a sharp increase up to $4{,}413 \pm 51$ yrs. BP in 0.7 m depth (Fig. 2 a). For the Red Sandstone soil profile there was only an increase in the ages from a modern like signature in 0.04 m to $532 \pm 41$ yrs. BP in 0.3 m depth. The Miocene Sand soil profile in 0.4 m depth showed an increase from $1{,}277 \pm 41$ to $1{,}771 \pm 44$ yrs. BP in 0.6 m depth. Thus, OC of the subsoil (around 0.6 m depth) in the Loess was more than twice as old as in the Miocene Sand. Contrastingly, the Loess has a modern like signature in 0.3 m while the soil developed in Red Sandstone showed an average age of $532 \pm 41$ yrs. BP in 0.3 m depth. This could be due to the observed plough layer at the Loess site mixing up the upper 30 cm with a predominantly modern $^{14}$C signature.

The ages of OC in the sediments ranged from 2,200-30,730 yrs. BP, with respective mean ages of $9{,}077 \pm 3{,}234$ yrs. BP for the Miocene Sand, $13{,}674 \pm 9{,}632$ yrs. BP for the Loess, and $14{,}463 \pm 1{,}992$ yrs. BP for the Red Sandstone. For all sediments, 11 out of 16 samples contained a $^{14}$C content that led to an apparent $^{14}$C age older than the soil age of 11,600 yrs. BP, assuming that soil development started after the latest glacial period at this time (Litt et al., 2007). Therefore sediments contain a mixture of geogenic ($^{14}$C free) and biogenic (with $^{14}$C) OC. Despite being the youngest sediment, the Loess partly revealed the highest apparent $^{14}$C ages with up to $30{,}730 \pm 631$ yrs. BP (Fig. 2 a). The ages of OC in the sediment of the Red Sandstone and the Miocene Sand ranged from $12{,}940 \pm 132$ to $17{,}390 \pm 206$ yrs. BP and from $6{,}750 \pm 86$ to $12{,}770 \pm 151$ yrs. BP, respectively revealing no depth trend with higher ages in the deeper sediment. The calculated GOC fraction in the sediments was highest for the Red Sandstone, ranging from 67 to 87 % with a mean of 77 % (Fig. 2 b). For the three samples of the Miocene Sand the GOC fraction ranges from 29 to 77 % with a mean of 53 %. The Loess showed a sharp increase at a depth of five metres where GOC contribution went up to 71 to 98 % while it was only 5 to 19 % in 2 to 4 m depth.

The calculated weight based content of GOC in the sediment revealed a comparatively uniform distribution for all sediments with depth, except the extremely high contents of the Loess in 5 m depth (Fig. 2 c). The investigated sediment depths revealed a quite narrow range of GOC contents with $0.10 \pm 0.03$ g kg$^{-1}$ for the Miocene Sand, $0.17 \pm 0.12$ g kg$^{-1}$ for the Red Sandstone and $0.27 \pm 0.08$ g kg$^{-1}$ for the Loess.

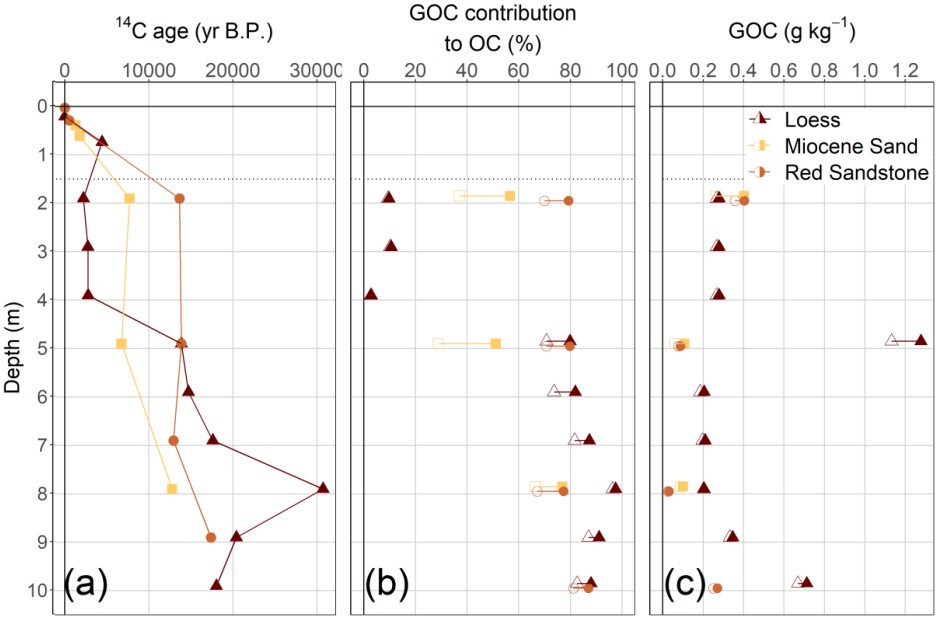

**Fig. 2:** Depth distribution of apparent $^{14}$C ages (in yrs. BP) (a), GOC contribution to OC contents in the sediments (b) and resulting weight based amounts of GOC (c) in the sediments. The range in the contribution from GOC is due to the assumption of an average biogenic OC age of 4,000 (empty shapes) and 1,000 years (filled shapes).

### 3.3 Biodegradability of sedimentary derived organic carbon

The incubation experiment revealed a potential but low biodegradability of OC for all samples, but without a clear depth gradient from 1 to 10 m (Fig. 3). For all substrates, the highest mineralisation for a stable OC pool occurred for the shallowest and the deepest depth increment (Table 2). While the Red Sandstone showed very low mineralisation when the samples were incubated as intact cores (0.03-0.9 mg $CO_2$-C g$^{-1}$ OC y$^{-1}$), the mineralisation rate constants were approximately four times higher when the samples were crushed (0.7-2.2 mg $CO_2$-C g$^{-1}$ OC y$^{-1}$) (Tab. S1). Comparing the two incubation experiments, the Miocene Sand showed an even higher increase in mineralisation rate constants (from 1.2-12 to 9.8-17.6 mg $CO_2$-C g$^{-1}$ OC y$^{-1}$) while the Loess revealed reduced mineralisation rate constants for three out of four samples (from 1-34.2 to 3.5-11 mg $CO_2$-C g$^{-1}$ OC y$^{-1}$). After 533 days, at the end of the second incubation experiment, the amount of mineralised OC decreased in the order Miocene Sand (0.79-1.52 %) > Loess (0.42-0.85 %) > Red Sandstone (0.17-0.42 %, Table 2). In contrast, the fitted models for the second incubation experiment with crushed Red Sandstone samples revealed that mineralisation rate constants were by far highest for the Red Sandstone. They were up to 100 times higher than respective mineralisation rate constants for the stable pool of the Loess and Miocene Sand. For the Loess and the Miocene Sand, mineralisation was to a large extent affected by the high mineralisation rates at the start of the incubation which is shown by the high mineralisation





rates of the labile OC fraction (Fig. 3, Table 2). Nevertheless, the two-pool model revealed that this
labile OC fraction  (*100–a* in Eq. 5) only represents an extremely low portion of mineralised OC (< 1
%). This resulted in calculated mean residence times for the stable OC pool of the Miocene Sand and
the Loess of > 10,000 yrs. compared to residence times < 1,000 yrs. for the Red Sandstone.

**Table 2:** Results from the incubation experiment: amount of mineralised OC within 533 days and the proportion,
mineralisation rate constant and mean residence time of the fast (*k1*) and slow (*k2*) degrading OC pool. Results
for the Red Sandstone are derived from the incubation experiment with crushed samples.

| Substrate | Depth (m) | Mineralised OC (% of total OC) | Labile OC | Stable OC | Labile OC | Stable OC | Labile OC | Stable OC |
|---|---|---|---|---|---|---|---|---|
| | | | | | Mineralisation rate (mg $CO_2$-C $g^{-1}$ OC $y^{-1}$)[a] | | Mean residence time (years)[b] | |
| Miocene Sand | 2 | 0.79 | 0.20 | 99.80 | 74 | 0.04 | 13 | 24,672 |
| | 4 | 1.01 | 0.06 | 99.94 | 127 | 0.07 | 8 | 15,381 |
| | 7 | 1.52 | 0.11 | 99.89 | 77 | 0.10 | 13 | 10,250 |
| | 9 | 1.06 | 0.54 | 99.46 | 35 | 0.04 | 29 | 27,742 |
| Loess | 1.7 | 0.45 | 0.04 | 99.96 | 200 | 0.03 | 5 | 35,148 |
| | 2.9 | 0.85 | 0.14 | 99.86 | 81 | 0.05 | 12 | 20,379 |
| | 5.4 | 0.78 | 0.09 | 99.91 | 146 | 0.05 | 7 | 21,038 |
| | 9.6 | 0.42 | 0.01 | 99.99 | 211 | 0.03 | 5 | 35,950 |
| Red Sandstone | 2.5 | 0.17 | - | | 1.15 | | 867 | |
| | 5 | 0.25 | - | | 1.69 | | 590 | |
| | 7.5 | 0.42 | - | | 2.88 | | 348 | |
| | 9 | 0.21 | - | | 1.43 | | 701 | |

[a] Derived from the mineralisation rate constant *k1* and *k2*
[b] Assuming a constant mineralisation with the rate constants *k1* and *k2*



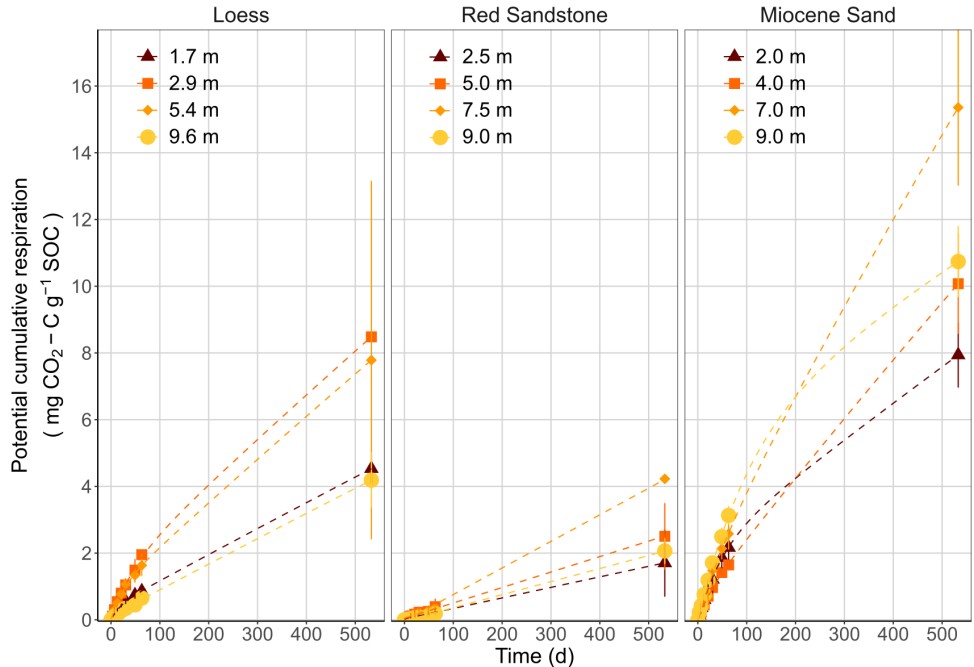

**Fig. 3:** Potential degradability of sedimentary OC from three sites. Results represent cumulative respiration from a 533 days incubation experiment at 20°C with respective standard deviations (n = 4). Dashed lines represent a fitted model to the respiration data. For the Loess and the Miocene Sand a non-linear regression model was fitted while for the Red Sandstone a linear model was used.

*3.4 Possible contribution from geogenic organic carbon to soil organic carbon*

When the bedrock is weathering it becomes part of the soil and also GOC becomes soil OC. The potential amount of GOC in the subsoil was dependent on the sedimentary bedrock (Fig. 4). In the subsoil of the Miocene Sand it added up to 2-3 %, at the Loess site it was 8 % and at the Red Sandstone site it was 12 % (Table 1). For the defined topsoils (0-30 cm depth), contributions of GOC to soil OC were smaller with 0.7-0.9 % for the Miocene Sand, 2.8-2.9 % for the Loess and 2.8-3.0 % for the Red Sandstone.

The presence of [14]C-free GOC to soil OC reduces the mean bulk soil OC [14]C ages depending on its proportion on soil OC content. Topsoils that developed in the Loess and the Red Sandstone had a modern [14]C content of 1.029 and 1.035 F[14]C similar to the atmospheric [14]C content in 1950. Because of the large proportion of biogenic OC, an influence of a geogenic fraction is not detectable in the topsoil of these sites. No [14]C data are available for the topsoil of the Miocene Sand. For all subsoils, influence of GOC on bulk soil OC [14]C contents depends strongly on the depth and the corresponding OC contents. For the Loess the possible influence on [14]C ages in the subsoils would be quite high, with





an average of 10 % reduction in mean apparent $^{14}$C ages in the subsoil. It would therefore reduce the
measured age from 4,413 yrs. BP in 74 cm depth by 532-555 yrs. BP. Geogenic OC potentially
reduces the mean apparent radiocarbon age of 1,277 yrs. BP in 0.39 m depth in the Miocene Sand by
about 7-9 yrs and the radiocarbon age of 1,771 yrs. BP in 0.61 m depth by 20-24 yrs.. This reduction is
below the respective standard deviations of the measurement. Nevertheless, in 1 m depth a given
possible proportion of 13.1-16.3 % would reduce an interpolated $^{14}$C age of 3,053 yrs. BP by 399-497
yrs. For the Red Sandstone the influence of GOC on $^{14}$C ages would be highest in the subsoil. In 74
cm depth it would influence an age of 1,453 yrs. BP by 451-490 yrs. BP. Due to the low amounts of
soil OC in 90 cm depth at the Red Sandstone site, the weight based median amount of GOC in the
sediments is even four times higher than the biogenic amount of soil OC.

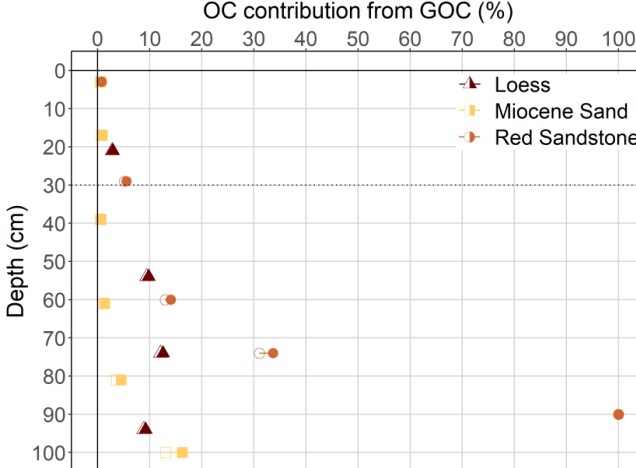


**Fig. 4:** Highest possible contribution from GOC to OC in relation to the bulk OC content. Here the median GOC contents of
the sediments were taken into account for the respective horizontal weight based OC contents. The range in the contribution
from GOC is due to the assumption of an average biogenic OC age of 4,000 (empty shapes) and 1,000 years (filled shapes).
The dotted line represents the defined boarder between top- and subsoils.





**4. Discussion**
*4.1 Geogenic organic carbon in the sediments*
Regarding our calculated contribution from GOC to OC in the sediments, the assumed range of
biogenic [14]C ages from 1,000-4,000 yrs. BP is within the typical range for ages of dissolved OC
leaching from soils (Artinger et al., 1996, Jia et al., 2019). Despite this, the range from 1,000-4,000
yrs. BP did not greatly influence the range of calculated sedimentary contribution from GOC,
especially for the Loess (~ 6 % difference) and the Red Sandstone (~ 9 % difference). The calculated
GOC contribution for the Miocene Sand was comparatively low (29-77 %) compared to the
contribution for the Red Sandstone and the Loess. This could be due to deep biogenic carbon inputs
e.g. as roots and root exudates from the trees (Angst et al., 2016, John et al., 2016, Tückmantel et al.,
2017, Kirfel et al., 2017), since the loosely bedded Miocene Sand allows for a deep infiltration
compared to the Red Sandstone site with its shallow bedrock as a root restricting layer (Schneider and
Don, 2019). Compared to the Loess site, the texture driven lower water conductivity (Saxton and
Rawls, 2006) could be responsible for lower contributions from biogenic OC. In the Loess, the low [14]C
ages in 2-4 m depth (2,200-2,770 yrs. BP) in contrast to the high [14]C age of 4,413 yrs. BP in 0.74 m
depth were surprising. This might indicate past anthropogenic activities or erosion driven material
movement that might have mixed up the upper part of the profile. Nevertheless, at depths below 4 m
the high [14]C ages of OC in the sediments indicated a large proportion of GOC. This could be due to
different sedimentation periods, soil forming and also soil burial processes (Chaopricha and Marin-
Spiotta, 2014) that took place during the Pleistocene. These processes can lead to the presence of
buried layers in the Loess with varying amounts of rather recalcitrant OC as it was shown by Marin-
Spiotta et al. (2014) and is indicated by the high variance in OC amounts within the Loess sediment.
In summary, the contribution from GOC to sedimentary OC was substrate dependent. A loosely
bedded sediment like the Miocene Sand with extremely low concentrations of OC is more prone for
infiltration of biogenic OC. This resulted in contributions of biogenic OC to the sediments of about 50
%. In contrast, the Loess site with comparatively low infiltration rates or the Red Sandstone site with
reduced possibilities for deep rooting seemed to contain relative constant contributions from GOC of
around 80 %.
We found that the GOC contribution within the sediments did not increase with increasing soil depth.
This is in contrast to the results of Frouz et al. (2011) showing that different sediment types from a
Miocene clay sediment had higher weight based carbon contents in 150 m compared to 30 m depth.
However, a comparison with the present study is rather difficult since Frouz et al. (2011) did not
distinguish between the geogenic and the biogenic OC fraction  and OC contents were by far higher
(28-112 g kg[-1] compared to 0.008-10 g kg[-1] in our results). But this also underlines the importance of





different sedimentation processes for the amount and depth distribution of OC in sediments and
sedimentary rocks.
A comparison of the weight based amounts of GOC in terrestrial sediments is difficult, since most
studies in that field rarely determine the amounts of OC in terrestrial sediments, or pre-assume that
e.g. sandy sediments contain no considerable amounts of OC (Artinger et al., 1996). Quite high
amounts of OC were found in the skeleton part of different soils on sandstones with 0.61-1.97 t ha$^{-1}$
Corti et al. (2002). Nevertheless, they mentioned the possible high influence of organic substances
from the soil solution without quantifying it and they did not directly investigated OC in the
sediments. Additionally Copard et al. (2007) assumed an OC amount of 2.4 g kg$^{-1}$ from an unknown
source for all sandy sediments in a global storage modelling approach for the first metre of the
sediments. This number is between 14-24 times higher than the median GOC amount in the Red
Sandstone (0.2 g kg$^{-1}$) and the Miocene Sand (0.1 g kg$^{-1}$). More data are thus needed to constrain the
OC pool in sedimentary bedrock since it will also influence the soil carbon pool in our study.
Loess deposits are comparably well investigated as archive for paleoenvironmental conditions (Wang
et al., 1996, Head et al., 1989, Hatté et al., 1998, Murton et al., 2015). The median amount of 0.27 g
kg$^{-1}$ from our study is low compared with studies from Hatté et al. (1998), Wang et al. (1996) and
Strauss et al. (2012). Hatté et al. (1998) investigated 20 m depth loess deposits in the Rhine valley and
found OC contents between 1.0-8.6 g kg$^{-1}$, Wang et al. (1996) investigated 12 m depth loess deposits
in China and found OC contents of $31.2 \pm 30.5$ g kg$^{-1}$ and Strauss et al. (2012) found OC contents of
$15 \pm 14$ g kg$^{-1}$ in Yedoma loess deposits in Siberia. This shows that the deposits from our investigated
site stored comparatively low GOC contents, although most of the studies mentioned above did not
distinguish between a biogenic and a geogenic OC pool. Nevertheless, also in our study Loess is,
compared to other sediments, a sedimentary bedrock with a high OC content. This was in line with
highest OC contents in subsoils at the Loess site and may indicate the importance and contribution of
bedrock OC to subsoil OC.
*4.2 Is sedimentary derived organic carbon biodegradable?*
The incubation experiment revealed a mineralisation of OC within the sediments with values between
0.02-0.3 % of total OC being mineralised after 63 days, and values between 0.2-1.5 % after 533 days.
A direct mineralisation of OC from sediments is in accordance with several studies investigating the
direct mineralisation from outcrops (Copard et al., 2007, Soulet et al., 2017, Horan et al., 2017, Petsch
et al., 2000). The difference to our study is, that they observed this mineralisation when the sediments
were directly exposed to the surface or/and part of a very fast eroding area. Thus GOC from the
sediments already is in touch with the atmosphere and inputs of the recent vegetation. However, Frouz
et al. (2011) conducted an incubation experiment with sedimentary samples from OC rich Miocene
clay sediments. They found quite high respiration rate constants with values between 3.5-12.3 mg



$CO_2$-C g$^{-1}$ OC y$^{-1}$. They attributed this to the prevailing presence of aliphatic compounds in their
samples being decomposed. Also Kieft and Rosacker (1991) found high respiration rates of
sedimentary samples with values between 0.9-9.5 mg $CO_2$-C g$^{-1}$ OC y$^{-1}$ which they primarily attributed
to the physiological status of the soil microbial community expressed as adenylate energy charge.
Since both studies were conducted on marine sediments with comparatively high OC concentrations of
0.52-11 g kg$^{-1}$, together with potentially different microbial community compositions, this might also
be a factor that drives the higher respiration (Colman and Schimel, 2013) compared to our samples.
Despite this Kieft and Rosacker (1991) did not distinguish between a fast and a slow degrading OC
pool. Compared to subsoil incubation experiments, the mineralisation found in our incubation
experiment was also quite low. For example, in subsoil incubation experiments at 20°C, Wordell-
Dietrich et al. (2017) found that between 0.5-0.95 % of OC are mineralised after incubation for 63
days, Wang et al. (2013) reported values between 0.5-1.5 % of after 28 days and Soucemarianadin et
al. (2018) reported values between 1-1.25 %  after 70 days. The difference is that our incubation
experiment lasted at least 5 times longer and the highest respiration took place at the beginning.
Comparing the modelled mineralisation rate constants for the stable OC pool from the two pool model
(0.028-0.049 mg $CO_2$-C g$^{-1}$ OC y$^{-1}$ for the Loess and 0.036-0.097 mg $CO_2$-C g$^{-1}$ OC y$^{-1}$  for the
Miocene Sand) shows that they are at least around 10 times lower compared to typical soil OC
mineralisation rate constants for a slow degrading OC pool with values between 0.4-11.7 mg $CO_2$-C g$^{-1}$ OC y$^{-1}$
OC y$^{-1}$ (Saidy et al., 2012, Rasmussen et al., 2006, Santos et al., 2012). Also the mineralisation rate
constants for the Red Sandstone (1.2-2.9 mg $CO_2$-C g$^{-1}$ OC y$^{-1}$) are within the lower range of these
rates. Thereby the comparatively high mineralisation rates for the Red Sandstone samples might only
be due to the duration of the incubation. Due to the mineralisation with time it seems not reasonable to
assume a higher mineralisation rate compared to the Miocene Sand and the Loess. Furthermore, the
low mineralisation rate of the Red Sandstone during the incubation as intact cores (Fig. S1) promotes
the stability of GOC when it is part of the sediments. This might be due to the low accessibility of OC
in the sediments for microorganisms and the low the availability of water due to a preferential flow
through the sandstone (Swanson et al., 2006). Altogether the low mineralisation rates of the OC in the
sediments might be due to the lack of fresh substrates and/or microorganisms that could enhance the
degradation of OC (Fontaine et al., 2007). Nevertheless, a mineralisation of OC could be observed
only by adding water, which indicates the presence of an active microbial community in the sediments
(Joergensen and Wichern, 2018, Magnabosco et al., 2018, Bomberg et al., 2017). Beside the observed
mineralisation, the extremely low calculated amounts of a labile pool (<0.22 % of total OC), according
to the fitted two pool model for the Loess and the Miocene Sand, clearly differentiates from the
biogenic OC pool (> 10 % of total OC) based on $^{14}$C measurements. Thus, there is no correspondence
between a fast mineralisable OC pool and the biogenic OC pool. This may be the result of a biogenic
OC pool in the sediments being mineralised and containing a readily bioavailable and degradable as



well as a more stabilised fraction. If the total bedrock OC would show mineralisation rates as observed
in our study, sediments still would become OC free within ≤ 36,000 years.
Taken together the long-term incubation cannot answer the question if GOC will be mineralised when
it becomes part of the (sub-)soil. The high residence times of the stable OC pool, however, indicate
that it might be relatively resistant against degradation when it becomes part of the subsoil. This is in
accordance to an indirect approach to determine the mineralisation of sedimentary OC when it
becomes part of the subsoil from Graz et al. (2010). They stated that 30 % of GOC resisted
degradation when it becomes part of the soil due to the results from a quantitative palynofacies
analysis of bedrock and soil samples. On the other hand Hemingway et al. (2018) found that
sedimentary OC directly exposed to the surface in a rapidly eroding tropical mountain area exhibits a
considerable mineralisation down to 1 m below the surface. Based on $^{14}$C measurements they found
out that on average $67 \pm 11$ % of the OC fraction in the sediments could be lost during soil formation
but did not distinguished between a biogenic and a geogenic OC fraction. This indicates that a
microbial mineralisation of bedrock OC takes place but may be partly restricted to biogenic OC.
Regarding the depth distribution of GOC in the sediments, the amount of GOC (in g kg$^{-1}$) does not
increase with depth but shows clear differences. On the one hand, this represents the sedimentation
history with different initial amounts of OC and degradation during sedimentation. This is particularly
evident by the high amounts of GOC in 5 m depth of the Loess. Meanwhile, contents of GOC
especially in the Red Sandstone and Miocene Sand are in the same range over the whole depth. This
might indicate that GOC is not degraded within the sediments. If there would be a degradation of GOC
within the sediments one would expect a decreasing amount with decreasing depth due to the input of
water, microorganisms and fresh nutrients from above. Furthermore, there is a relatively constant
contribution from biogenic OC within the sediments. This means that, if biogenic OC enters the
sediments, together with possibly degrading microorganisms, this biogenic OC might also be
preferably mineralised. This gives a hint that GOC is not degradable when it becomes part of the (sub-
)soil, since especially for the Loess and the Miocene Sand, the conditions within the sediments in > 1.5
m depth do not differ so much from the conditions (e.g. oxygen and water content) within the subsoil.

*4.3 How much GOC contributes to soil organic carbon?*

The contribution of GOC to soil OC stocks in our study is driven by the amount of OC in the soil and
the amount of GOC in the respective sediment. Our results revealed that despite differences between
sediments, GOC content varied in a quite narrow range between 0.1 and 0.3 g kg$^{-1}$. The contribution of
GOC to topsoil OC was negligible. Highest possible contributions of GOC to total subsoil OC were
found for the Red Sandstone (~30 %) and lowest for the Miocene Sand (0.6 %). This was due to the
range of OC contents in the subsoils (0.53 g kg$^{-1}$-15.21 g kg$^{-1}$). When soil OC contents were low, the
possible contributions from GOC were high and vice versa (Fig. S2). For our investigated soils OC



contents of 3 g kg$^{-1}$ soil allowed for possible GOC contributions between 5-10 %. For OC
around 1 g kg$^{-1}$ soil a GOC contribution between 10-20 % seems to be possible. Thereby higher
contributions came from GOC rich sediments like the Loess and lower contributions from sandy
sediments. In comparison, van der Voort et al. (2018) estimated the contribution from GOC of a soil
derived from glacial deposits (flysch) between 80-100 cm depth to be around 40 %. For a soil
developed from a poorly consolidated sedimentary rock (calcareous and shaly moraine) they
calculated the contribution from GOC to range from 20 % in 145 cm depth to 80 % in 310 cm depth.
There has further been an attempt to fractionate subsoils to extract the most stable OC that may be
derived from GOC. Paul et al. (2001) found that 30 % of subsoil OC was non-hydrolysable. The
investigated a soil developed on loess over till with this non-hydrolysable fraction showing a $^{14}$C age
of 13,000 yrs. BP. They also concluded that this high age can partly be explained by a GOC fraction.
These results indicate that especially deposits from the past glacial periods like flysch or till have a
much higher potential for OC contributions from GOC possibly due to their higher amounts of GOC in
the sediments. Since we only investigated terrestrial sediments it has to be taken into account that also
marine sediments or shales contain much higher amounts of OC up to 250 g kg$^{-1}$ (Hemingway et al.,
2018, Petsch et al., 2000). Their amount of GOC and possible contribution to subsoil OC stocks might
therefore be much higher.
Nevertheless, $^{14}$C ages of OC in the subsoil can also be high in soils derived from igneous parent
materials without GOC (Rumpel et al., 2002). Furthermore, on a global scale $^{14}$C ages of soil OC are
primarily driven by climatic conditions, clay content and age, since soil development started (Mathieu
et al., 2015). But for terrestrial sediments with comparatively low amounts of GOC that started their
soil development after the latest glacial period, we could obtain a scale for possible contributions when
the amount of OC is known. Thus, at a global scale the high $^{14}$C age of subsoils is not only driven by
the GOC fraction but the presence of GOC may considerable influence subsoil $^{14}$C.
**5. Conclusion**
With our approach of estimating the GOC contribution to soil OC, we could show that common and
abundant terrestrial sediments, with low amounts of sedimentary OC, can contribute considerably to
subsoil OC stocks. One fraction of OC in the sediments is of geogenic origin and could therefore
influence measured $^{14}$C ages in soil, in particular in subsoils. Subsoils are known for their high $^{14}$C
ages and slow turnover rates and slow reaction to changing environmental condition. These properties
of subsoil OC may partly be derived from the GOC in the subsoil. The sediments at the investigated
sites contained OC in a range from 0.1-0.3 g kg$^{-1}$, allowing for contributions from GOC between 10-30
% in subsoils. We have also shown that this geogenic contribution presents a quite stable OC pool,
especially for subsoils. Thus, also sediments with comparatively low amounts of OC, could show
considerable contributions from GOC.



**Data availability**

The data will be made available on request

**Author contribution**

AD conceived of and designed the study, FK performed the sampling and analysis, and wrote the first draft. All the authors contributed to generating and reviewing the subsequent versions of the manuscript.

**Competing interest**

The authors declare that they have no conflict of interest

**Acknowledgements**

This study was funded by the Deutsche Forschungsgemeinschaft (DFG) (DO1734/4-2) within the framework of the research unit SUBSOM (FOR1806)-"The Forgotten Part of Carbon Cycling: Organic Matter Storage and Turnover in Subsoils". We would like to thank Frank Hegewald and the student assistants for their support in the field and in the laboratory, Jens Dyckmanns and Lars Szwec from the Centre for Stable Isotope Research and Analysis at the University of Göttingen for $^{13}CO_2$ measurements and Reiner Dohrmann from the Federal Institute for Geosciences and Natural Resources for density measurements of sedimentary rocks. We would also like to thank the AK laboratory team for their support.





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
