# Peer review of "Geogenic organic carbon in terrestrial sediments and its"

_SOIL, 2020_

## Referee Comment (RC1) · Anonymous Referee #1 · 16 Aug 2020

Review Kalks et al. 2020

"Geogenic organic carbon in terrestrial sediments and its contribution to total soil carbon" by Kalks et al. is a well written, well structured and timely contribution to an interesting topic, namely the varying contribution of geogenic C to the terrestrial C cycle. For this the authors study C dynamics in a depth explicit way on a range of sediment cores taken from different sources of parent material for soil development in central Germany. It's a good manuscript that falls into the scope of the journal.

My main comment is the lack of confidence that the authors provide at this point in the several of the analytical measurement done and in the statistics behind it. The apparent low number and/or absence of replication for some of the analyses worries me a bit too. Especially as there are some datapoints that were discussed including mechanistic

interpretation (for example, inorganic C variability) that look more like outliers. I give more detail on this below and I am sure the authors can address this as part of the revision process.

Methods: 1. Provide Info on soil depths and weathering depths. It should be rather easy to see from the cores where soils start and end to give at least a relative indication of difference in soil depth between the three cores. This is important as you argue later on with variable C inputs which should have an impact on weathering.

2. Clarify – How has the sample been chosen for each depth increment that was later analyzed and treated? Is this a composite of each 1m increment, taken at the center of the increment etc.

3. Due to the very low carbon concentrations that we have been measured in these sedimentary rocks, giving confidence in the reliability of the measurements is extra important. For example, 1M HCL was used for decarbonatization of samples for 14C analyses, but for the rest inorganic C was assessed using loss on ignition parallel to dry combustion for total CN. Can you say something on the uncertainty of the methods, detection limit and replication for loss on ignition vs total combustion vs acid hydrolysis? I assume the uncertainty varies considerably, varies with depth and concentration and might related due to incomplete and variable assessment of inorganic C which would affect a number of conclusions

4. One thing that concerned me with the methods was that Loess deposits seem to be free of inorganic C -which shouldn't be the case for unweathered deposits- except for some spikes at greater depth.

5. During the incubation, have any amendments except keeping water constant been made? 533 days is a long time without additions and microbial activity will be affected. You state in your discussion that you expect some C input through exudates to play a role, so this would be something to consider when interpreting your respiration.

6. How have you treated problems of oversaturation with $CO_2$ if the containers were packed air-tight or have they been flushed with ambient air except for a shorter time before $CO_2$ measurements to accumulate gas for sampling? There would be gas exchange in nature in these sediments and standard deviation in Figure 3 reveals quite high variability for the 533 day sampling points, especially for loess. And if they were gas tight, why not analyzing $CH_4$, which is more important in oxygen deprived environments.

7. How many replicates have been used during incubation?

8. Give an overview on these mineralization rate constants that you took from Qualls and Haines. As the authors know, a lot has happened since then in terms of re-defining pool models and C turnover and the Qualls and Haines study was on dissolved organic carbon and turnover there. I think you need to provide some confidence why the rates and equations provided there are applicable for such a different system as your soil/powdered rocks experiment. Given the uncertainty surrounding the assumptions behind the pools I wonder if it won't be better to leave out the pool model altogether and just work with observed data assuming a linear trend of respiration between two measurement points along the timeline. I believe 14C measurements on the respired C across the length of the incubation experiment would have helped.

9. For the analysis of how much C has been mineralized, were soil and sediment samples measured after the incubation again to check if your $CO_2$ loss calculations and mineralization rates make sense?

10. What confidence can you give for the $CO_2$ respiration assessment between days 63 and 533? Figure 3 shows that the curves differ a lot between those two phases of the experiment.

11. Stats: I could not follow the authors argument why standard error and significance could not be displayed in the manuscript. Yes, the model output might be tricky, but other measured and experimentally assessed parameters can and should be displayed

with some statistical confidence to avoid speculating on outliers in the depth trends of the data.

Results and Discussion: 12. You might want to give the discussion some more sub-headers. For example, l.429-439 vs the section before and after seem to be distinct from each other. Will help to structure it

13. Parts of the discussion are speculative. Here the examples I would see some revision on:

14. Roots and root exudates have been named for deep biogenic C inputs. Name the depths you are referring to for this and provide some estimates on rooting depth if available. As your data does not show strong depth trends for >4m soil depth, which would be expected if C cycling is still tied to DOC inputs

15. What evidence do you have to expect soil burial and soil formation during the Pleistocene avoiding a circular argument with GOC as an indicator? I am not sure what to make of this argument.

16. You need to discuss the fact that you incubated at 20°C, whereas temperature at the depths in which these sediments reside will be at the mean annual temperature of the study region. So roughly 9°C. That's a giant step in terms of potential energy available for microorganisms.

17. I think some discussion on the quite varying depositional regime between the three geologies is necessary as part of the discussion on why sedimentary C is bioavailable or not. Some of the more degradable components might be lost before sedimentation and overpower variability in biodegradability compared to stabilization of C in soils and sediments.

18. L. 550-556 Seems disconnected to me from the rest of the discussion. Consider deletion

Further comments:

The title is a bit misleading. I think it would be better to say "contribution to terrestrial (or soil) carbon cycling" – as the study also involves incubations and isotope work and not just stocks or similar as indicated at this point.

pH measured with what? H2O, KCl, CaCl2? - Specify in methods

The text is well written, but there are shortcomings in wording and grammar all over the manuscript. Its nothing that stops the reader from following, but I suggest a native English speaking colleague checks this manuscript before submission.

Some examples (I only picked a few, but there are more): l. 135. Grammer: "heated" instead of "heating". l. 214. Pouring "bulk" density is the correct term I believe. l.441 median of 0.27g kg-1 of what? l.542. Check grammar.

---

## Referee Comment (RC2) · Anonymous Referee #2 · 27 Aug 2020

The paper on "Geogenic organic carbon in terrestrial sediments and its contributions to total soil carbon" by Fabian Kalks et al. is a very interesting contribution, and fits well with the journal topics. It is a good paper, but the authors are omitting the possibility that a fraction of the geogenic organic carbon (GOC) in deep soil can be assimilated by bacteria. This has been shown to occur in district environments such as in very isolate environments (please cite (Schwab et al. 2019)) or in deep soil (please also (Seifert et al. 2011)). Bacterial assimilation of GOC will generate 14C-dead easily accessible C (here labile pool), which can easily become a part of C dynamic in soil. This has important implications on the author discussion/conclusion (e.g. age of the soil, model) and needs more thorough discussion. Based on the results of the incubation experiments, the authors concluded that GOC mineralization and thus bacterial assimilation is very low or insignificant. However, as already noted by the first referee (point 5), "the incubation of 533 days is a long time without additions". In deep soils, inputs from surface are expected to fuel bacterial activity which would favor the mineralization and the assimilation of GOC. Incubation without addition most likely underestimated mineralization rate (bacterial assimilation) of GOC in soil. More emphasis has to be put on this in the manuscript. I give more detail on this below.

Introduction: Bacteria fixing GOC will produce fresh 14C-depleted organic matter. The method mentioned here will thus overestimate GOC.

Method: More details on land-use history of the sampled soils are needed. What about the soil samples from agriculture field? What about the effects of the different vegetation/land-use history on soil weathering, DOC, root input? What about O2 during the incubation? It is difficult to follow the soil treatments before incubations. Why only show the incubations with the intact rocks in Fig.3, as crushed rocks better estimated the effect of weathering and showed significant higher respiration rates.

Line 203: please delete under optimal conditions.

It is important to mention that the calculation (eq. 7) assumes that no labile faction is derived from 14C-dead bacterial biomass.

Discussion Line 495-499: This is again assuming no bacterial assimilation. Please discuss this. Also, delete the last sentence line 499, as this would argue the opposite Line 512-524: same as before. 14C values of the respired CO2 could help here.

Conclusion: Line 565: change to" Incubation of sediments seem to indicate that this geogenic contribution. . .

Schwab, V.F., M.E. Nowak, C.D. Elder, S.E. Trumbore, X. Xu, G. Gleixner, R. Lehmann, G. Pohnert, J. Muhr, K. Küsel and K.U. Totsche. 2019. 14C-Free Carbon Is a Major Contributor to Cellular Biomass in Geochemically Distinct Groundwater of Shallow Sedimentary Bedrock Aquifers. Water Resources Research. 55:2104-2121. Seifert,

[Figure]

A.-G., S. Trumbore, X. Xu, D. Zhang, E. Kothe and G. Gleixner. 2011. Variable effects of labile carbon on the carbon use of different microbial groups in black slate degradation. Geochimica et Cosmochimica Acta. 75:2557-2570.

---

## Author Comment (AC1) · 25 Sep 2020

We thank the reviewer #1 for the very helpful comments to our manuscript. We will take all comments into account for a revised version of the manuscript. See below our comments to the issues raised by the reviewer.

Methods 1. Provide Info on soil depths and weathering depths. It should be rather easy to see from the cores where soils start and end to give at least a relative indication of difference in soil depth between the three cores. This is important as you argue later on with variable C inputs which should have an impact on weathering. - We will add more detailed information about the soil characteristics and weathering depths based on soil profiles. For the purpose of comparability of OC stocks we will keep the subsoil

definition for 150 cm depth.

2. Clarify – How has the sample been chosen for each depth increment that was later analyzed and treated? Is this a composite of each 1m increment, taken at the centre of the increment etc. - We will add information on soil sampling for chemical analysis: samples have been taken from a depth of $\sim$ 90 cm of each 100 cm increment (so 1.9 m, 2.9 m, 3.9 m etc). This decision was based on two aspects: First we wanted to have the same depth for each material. Second, Loess samples were obtained in closed cylinders that could only be opened at the top or bottom. To avoid taking samples from possibly contaminated edge of the cylinders we removed the first 5 cm and also removed the outer material inside the cores (see l. 129).

3. Due to the very low carbon concentrations that we have been measured in these sedimentary rocks, giving confidence in the reliability of the measurements is extra important. For example, 1M HCL was used for decarbonatization of samples for 14Canalyses, but for the rest inorganic C was assessed using loss on ignition parallel to dry combustion for total CN. Can you say something on the uncertainty of the methods, detection limit and replication for loss on ignition vs total combustion vs acid hydrolysis?I assume the uncertainty varies considerably, varies with depth and concentration and might related due to incomplete and variable assessment of inorganic C which would affect a number of conclusions - We will add information about the uncertainty of the method. We measured carbon contents per loss on ignition because we wanted to know organic and inorganic contents and not just the organic carbon content (that we would have received if we used decarbonised 14C samples). The uncertainty arised from the measured values and repetitions. Lowest measured values of total C in the cores samples (n = 3) was found in sand samples (0.04 mg C g-1 soil) while the same samples showed C values of 0.00 and 0.01 mg g-1 soil after the OC was removed at 450°C. Because this variation between 0.00 and 0.01 was random between the replicates, we assumed sand samples to be free of inorganic carbon. Furthermore 0.01 mg C g-1 soil can be seen as the detection limit of the used ignition method and 0.03 mg

OC g-1 therefore as a reliable content. For 14C measurements, respective samples always were decarbonised.

4. One thing that concerned me with the methods was that Loess deposits seem to befree of inorganic C -which shouldn't be the case for unweathered deposits- except for some spikes at greater depth. - According to Wagner (2011) ("Spatial analysis of loess and loess-like sediments in the Weser-Aller catchment, Lower Saxony and Northern Hesse, NW German") the investigated Loess at the Ahlshausen site belongs to the Leine Ihlme Basin and is referred to as "loamy loess" or "loess loam" that has been decalcified during weathering and soil genesis. We will add this information in the result section.

5. During the incubation, have any amendments except keeping water constant been made? 533 days is a long time without additions and microbial activity will be affected. You state in your discussion that you expect some C input through exudates to play a role, so this would be something to consider when interpreting your respiration. - During the incubation no further amendments have been made – they were completely sealed. Although we considered the fact that we did not added substrate during the incubation in the discussion (as we observed mineralisation only by adding water) this will be discussed more detailed.

6. How have you treated problems of oversaturation with CO2 if the containers were packed air-tight or have they been flushed with ambient air except for a shorter time before CO2 measurements to accumulate gas for sampling? There would be gas exchange in nature in these sediments and standard deviation in Figure 3 reveals quite high variability for the 533 day sampling points, especially for loess. And if they were gas tight, why not analyzing CH4, which is more important in oxygen deprived environments. - The containers have been kept air tight for the whole period of incubation. The air volume of the containers was 5 l and due to the comparatively low production of CO2 we can assume that no oxygen limitations occurred. We further had no problems with oversaturation, since total CO2 values stayed within the calibration range (maximum value was 7650 ppm) and concentrations did not exceed typical concentrations of $CO_2$ in subsoils. We also analysed all samples for $CH_4$ concentration which were on a very low level. Oxygen concentrations were all the time >20 Vol%. We will add this information in a supplement graphic. Thus, there were no indications for oxygen limited conditions.

7. How many replicates have been used during incubation? - We used 4 replications for the incubation. This is mentioned in the material and methods section (l. 206 in chapter 2.4) and also in the caption from Fig. 3. Since some few samples were not air tight at the end and were removed, especially after 533 days, which reduced the number of repetitions.

8. Give an overview on these mineralization rate constants that you took from Qualls and Haines. As the authors know, a lot has happened since then in terms of re-defining pool models and C turnover and the Qualls and Haines study was on dissolved organic carbon and turnover there. I think you need to provide some confidence why the rates and equations provided there are applicable for such a different system as your soil/powdered rocks experiment. Given the uncertainty surrounding the assumptions behind the pools I wonder if it won't be better to leave out the pool model altogether and just work with observed data assuming a linear trend of respiration between two measurement points along the timeline. I believe 14C measurements on the respired C across the length of the incubation experiment would have helped. - We used a two pool model because we assumed that mineralisation behaviour can be explained by the degradation of a labile and a stable OC pool. However, measurements after 533 days are less reliable (due to removed non-air tied samples) and there was a large time lag between sampling at day 63 and at day 533 days indeed. We will take the reviewer comment into account and apply a linear respiration model. Additionally, we will leave out the sampling point after 533 days and focus on the comparison between the first and the second incubation experiment (graphics will be changed and results and discussion applied). 14C measurements might have helped and have been intensively

discussed during preparation of the incubation experiment. But due to several technical limitations while working with sedimentary samples (e.g. extremely low CO2 production from the sedimentary samples leading to the problem of not detectable changes in the 14C signal) we could not apply them. The background CO2 in the bedrock material could hardly be removed in order to start with a CO2 free incubation.

9. For the analysis of how much C has been mineralized, were soil and sediment samples measured after the incubation again to check if your CO2 loss calculation sand mineralization rates make sense? - No, we did not measured OC concentrations after the incubation experiment. The incubation indicated that roughly 1 % of OC was lost due to mineralisation. Having a mean relative standard deviation of 13 % (analytical error derived from laboratory replicates) for the sand samples we do not think that measure samples after incubation will lead to results that could allow for a verification. Furthermore the blank samples from the incubation showed no trend of increasing CO2 (in fact the concentrations slightly decreased within the 533 days) so that we can exclude a possible contamination being responsible for the observed mineralisation.

10. What confidence can you give for the CO2 respiration assessment between days 63 and 533? Figure 3 shows that the curves differ a lot between those two phases of the experiment. - See comment on 8. We will restrict the main assessment to the first 63 days and we will show the respiration behaviour for the whole 533 day period in the supplement for each single sample without grouping them.

11. Stats: I could not follow the authors argument why standard error and significance could not be displayed in the manuscript. Yes, the model output might be tricky, but other measured and experimentally assessed parameters can and should be displayed with some statistical confidence to avoid speculating on outliers in the depth trends of the data. - We will add the mean relative standard deviation from the laboratory replicates for each sample. Since we have only two field replicates and one Loess core we cannot use standard deviations from field replicates., We further will add results from the two cores in Fig. 1 to show the variation for the OC, D13C and IC content

derived from two cores.

Results and Discussion

12. You might want to give the discussion some more sub-headers. For example, l.429-439 vs the section before and after seem to be distinct from each other. Will help to structure it - We will add subheaders for section 4.1 of the discussion

Parts of the discussion are speculative. Here the examples I would see some revision on: 14. Roots and root exudates have been named for deep biogenic C inputs. Name the depths you are referring to for this and provide some estimates on rooting depth if available. As your data does not show strong depth trends for >4m soil depth, which would be expected if C cycling is still tied to DOC inputs - We will add information on possible rooting depths for the sites and the implications for the observed trends in the discussion. In particular in sandy substrate with tree vegetation rooting depth can very deep.

15. What evidence do you have to expect soil burial and soil formation during the Pleistocene avoiding a circular argument with GOC as an indicator? I am not sure what to make of this argument. - We added this information to explain the discrepancies between bulk parameters and incubation results from different depths of the Loess sample. The reason to expect soil burial and soil formation is that we observed different layers within the loess sediment in terms of colour and OC contents. These layers formed during different climatic periods (warmer, cooler) as described in detail by Jordan & Schwartau (1993) who investigated the loess site and could assign the different layers to specific sedimentation periods during the Pleistocene.

16. You need to discuss the fact that you incubated at 20âŮęC, whereas temperature at the depths in which these sediments reside will be at the mean annual temperature of the study region. So roughly 9âŮęC. That's a giant step in terms of potential energy available for microorganisms. - The incubation should not simulate degradation under real environmental conditions but the potential biodegradability and to make it comparable to other incubation results. We will add explanation about the biodegradability of GOC in dependence of temperature regarding the different Q10 values in the discussion section. In the material and methods section we will add an explanation why we chose 20°C.

17. I think some discussion on the quite varying depositional regime between the three geologies is necessary as part of the discussion on why sedimentary C is bioavailable or not. Some of the more degradable components might be lost before sedimentation and overpower variability in biodegradability compared to stabilization of C in soils and sediments. - We will add information about the quality of OC in the sediments (unpublished work about the quality of GOC in the sediments) and the different microbial communities that could be derived from the sediments and their potential respiration rates (e.g. the paper from Fredrickson & Balkwill 2006) in the discussion section 4.2.

18. L. 550-556 Seems disconnected to me from the rest of the discussion. Consider deletion - This part is indeed somehow disconnected from the rest of discussion in this section, but we think it is important to draw attention to the situation of having igneous parent materials and a global classification of our findings. Therefore we will keep and revise it and we will add a new header for this small section.

Further comments 19. The title is a bit misleading. I think it would be better to say "contribution to terrestrial(or soil) carbon cycling" – as the study also involves incubations and isotope work and not just stocks or similar as indicated at this point. - This would move the focus of our study from the contribution and stability part towards a global carbon cycling view which we think is too broad for our experimental design. The focus of the study should remain on the contribution of GOC in subsoils.

20. pH measured with what? H2O, KCl, CaCl2? - Specify in methods - Measured with 0.01 mol L-1 CaCl. Will be specified in the method section.

21. The text is well written, but there are shortcomings in wording and grammar all over the manuscript. Its nothing that stops the reader from following, but I suggest a

native English speaking colleague checks this manuscript before submission. Some examples (I only picked a few, but there are more): l. 135. Grammer: "heated" instead of "heating". l. 214. Pouring "bulk" density is the correct term I believe. l.441median of 0.27g kg-1 of what? l.542. Check gramma - We will again check for these grammar mistakes. However, this manuscript was already checked by a native English speaker and professional language editor.

---

## Author Comment (AC2) · 25 Sep 2020

We gratefully thank the reviewer #2 for all the comments and advises to improve the manuscript. We will take all comments into account in a revised manuscript version.

22. Bacteria fixing GOC will produce fresh 14C-depleted organic matter. The method mentioned here will thus overestimate GOC. - The organic matter taken up by bacteria is the material in the place where they live. Thus, even if bacteria produce fresh 14C-depleted organic matter, its origin is still geogenic, and will thus not overestimate GOC. We will change the sentence in l. 55 to "OC that originates from deposition during sedimentation and rock formation" to make it more clear. Methods

23. More details on land-use history of the sampled soils are needed. What about

the soil samples from agriculture iňĄeld? - We will add more information about the sites from a detailed soil classification. In addition, we will give an estimation about the duration of historical agricultural use of the Loess site.

25. What about the effects of the different vegetation/land-use history on soil weathering, DOC, root input? - The different effect of vegetation/land-use history on past soil weathering can be neglected for the samples sites to the best of our knowledge since it is more the initial substrate and the atmospheric input that drives the weathering rates according to Watasuki (1992, "Rates of weathering and soil formation"). However, the effect of different land use on DOC input and deeper rooting trees will be discussed in more detail.

26. What about O2 during the incubation? It is difiňĄcult to follow the soil treatments before incubations. - Since the vessels were flushed with ambient air before the incubation started (see l. 217) and the air volume was quite high (around 5 l) we can assume that the O2 is not limiting during the incubation but O2 concentrations were always >20Vol%.. We will describe the soil treatments before incubation more detailed.

27. Why only show the incubations with the intact rocks in Fig.3, as crushed rocks better estimated the effect of weathering and showed significant higher respiration rates. - Fig. 3 shows the respiration rates for the crushed samples and not for the intact cores. We will make this clear in the Figure caption and in the text.

28. Line 203: please delete under optimal conditions. - Will be deleted

29. It is important to mention that the calculation (eq. 7) assumes that no labile faction is derived from 14C-dead bacterial biomass. - Since we will replace the double pool model (see comment on 8) and replace it with assuming a linear mineralisation model, we will rewrite it. 14-C dead microbial biomass as part of a possible labile OC fraction in the sediments will be discussed.

30. Discussion Line 495-499: This is again assuming no bacterial assimilation. Please

discuss this. Also, delete the last sentence line 499, as this would argue the opposite

Line 512-524: same as before. 14C values of the respired CO2 could help here. - This will be changed because we will replace the double pool model by linear regression analyses. But as responded to 29, we will now also discuss the possible microbial contribution with 14C free biomass. Since we do not have 14CO2 values (see comment on 8) we have to work with the data we obtained.

31 Line 565: change to" Incubation of sediments seem to indicate that this geogenic contribution. . . - Will be changed. Additionally - The suggested literature (Seifert et al. 2011 and Schwab et al. 2019) will be cited.

---

## Referee Report (RR1)

**Review of manuscript "Geogenic organic carbon in terrestrial sediments and its contribution to total soil carbon" of Fabian Kalks et al.**

I read the manuscript of Kalks et al. with great interest. It is a good study that brings the importance of geogenic derived organic C in (sub)soil carbon stocks to attention and increases our understanding of its (potential) role in soil C cycling. As such, it fits well the topic of the Soil journal.

However, despite the points addressed already by the previous reviewers, there are still some things to be cleared up. Especially the part on $\delta^{13}C$ is only very shortly mentioned in the introduction, without explanation or reference, and not at all used in the discussion section. These results have to be better integrated in the paper.

Furthermore, it is better to stick with geogenic organic carbon (GOC) as term and avoid using other terminologies like "sedimentary OC" or "Sedimentary contribution" as it causes some confusion. For example, it is not very clear if OC of section 3.1 was already corrected for GOC contribution. A better distinguish between OC and "corrected OC" has to be made, if this is the case.

Here below I gave more details on the issues encountered.

Major points to be addressed:

- Is there something known about the origin of GOC in the three different sites (i.e. buried organic matter / soot / coal / …)? As it is not stated/hypothesized, it makes also difficult to evaluate the used temperatures (450 ℃) for preheating samples to remove OC.

- Line 54: "GOC in most cases is devoid of 14C and thus may lead to an overestimation of ancient OC sources although a number of studies showed the importance of root derived, young OC inputs to subsoils." This is quite a fundamental point of your study and could be better stated here already. Suggested: "As 14C has a half lifetime of 5730 years, carbon deposited from the Weichselian and older are depleted in 14C, thereby diluting the overall 14C concentration. Especially in C poor subsoil, where GOC forms a relative larger part of the overall C content, this leads to an age overestimation of relative fresh OM, like root derived components. "

- Line 80: "Thus, using both carbon isotopes can reveal if the OC is a mixture of GOC and OC". It is not clear how $\delta^{13}C$ can be used (from the introduction) and more detail how these different isotopes can be used to disentangle the different C components should be added. Above this line it is only made clear why GOC and 14C are important to study.

- Line 101-102: Restructure and rephrase questions, especially as question II is fundamental for the disentanglement of geogenic and more recent OC. It is suggested to start with "Is (G)OC free of 14C", than "how much does GOC contribute to (sub)soil OC?" and "will  GOC be degraded and/or incorporated in recent OC"

- Line 261-266: This part of the results does not create confidence in your data. First it is stated all samples were within detection limit (to my opinion an understatement, as otherwise samples should not be included or represented by the value 0) and next there is speaking of "random noise". Better to simply state what the mean relative standard deviation was (or overall measurement/methodological error) and the lowest measured value (0.04 **g C kg$^{-1}$ soil**). Note that mg C g-1 soil and g C kg-1 soil are both used in the text.

- You could consider to discuss first the "How much GOC contributes to soil organic carbon?" before going into the bioavailability of it. This would make the "flow" of the discussion more logical.

Minor suggestions for improvement:

Line 22: "this gap" -> this knowledge gap

Line 24: "sedimentary OC" -> GOC

Line 51: "an contribution" -> "a contribution"

Line 60: "have been investigated" Missing the results of these studies, probably rephrase.

Line 68-69: "more information about the amounts of OC in sediments is needed." -> "GOC in sediments" or "contribution of GOC in sediments"

Line 74: "hydraulic conductivity" -> "Pore distribution" or "porosity" fits the context better.

Line 127-128: "This means e.g. for a sample increment from 1-2 m, the sample represents the 1.85-1.95 m depth" -> "This means that for example the increment 1-2m is represented by a sample from 1.85-1.95m depth."

Line 148: "removing carbonates" Same as 14C?

Line 250: "*lm*" -> "the function *lm*"

Title 3.1: "…sedimentary and subsoil organic carbon" -> be consistent with terminology. Better to use GOC / geogenic organic C instead of "sedimentary", especially as "subsoil organic carbon" can be all OC found in the subsoil

Line 296/277: Fig 1a. -> Fig 2a.

Line 284 "they all were in the range of C3 plant material. A value above -25 ‰ for the Red Sandstone in 4 m depth can be explained by corresponding high values of inorganic carbon (IC) in this depth" -> better for discussion

Line 308: "Fig. 2 a" -> "Fig. 3a"

Line 322: "Fig. 2 c" -> "Fig. 3c"

Line 424: "the same site assigned the different" -> "the same site **and** assigned the different"

Line 425: "sedimentary OC" -> "OC"

Line 426/427: "…extremely low concentrations of 426 OC is more prone for infiltration of biogenic OC" Not completely clear what is meant, but probably best to say:"… very low OC contents increases the relative importance of biogenic C input for the over OC"

Line 463: "sedimentary bedrock" Loess is no bedrock, but an (aeolian) deposit or sediment

Line 523: Not clear what is meant with "a resistant part"

Line 525: "distinguished" -> "Distinguish"

Line 526: "bedrock OC" -> "GOC"

Line 557: "despite differences between sediments" -> ", despite differing between sediments,"

Line 570-571: Combine sentences

Line 572: "high age" -> "high $^{14}$C age"

---

## Author Response (AR2)

Dear editors and reviewers,

We are very grateful for all your comments that helped us to revise and improve our manuscript. We took all comments into account. We also used the help of a professional language editor to get rid of language mistakes.

We hope that you agree with us that the manuscript improved in a way that it can be published in SOIL. We think that this is an important study to raise the awareness of sediments as understudied source for soil carbon.

Best regards

Axel and Fabian on behalf of all co-authors

**Revision reply 2**

**Editor comment**

I think that your determination of the GOC contribution at depth is highly questionable for two reasons:
1) the artificial setting of the border of transistion from soil to sediment at 1.5 m (why didn'y you use the 14C measurements for this?)

-Thank you for pointing to this issue that require claification. The transition setting at 1.5 m has no effect on our calculated contribution of GOC at depth. The calculation of GOC was based on $^{14}$C ages in the sediment cores (starting at 1.9 m depth). There is no clear boarder between soil and sediment. During soil sampling we still found pedogenic influence down to maximum depth of the soil profile. That is why we choosed to assume a boarder for the soil-sediment transition at 1.5 m depth. Please not, this transition was only used to compare **OC stocks** in the discussion. In l. 175 we stated that this transition was used to compare contributions from GOC. This may be confusing because thats only true for stocks and will be revised.

2) the use of literature values for 14C ages for DOC from subsoil for the quantification of GOC - this is highly uncertain - firstly 14C ages of DOC do not represent the SOC stored in solid form and second, literature values may not correspond to the study site It might have been better to use the youngest and oldest age possible for biogenic SOC (i.e. recent to 10000 years) to express the uncertainty of the results

- Yes, we agree with you in that this is a source of uncertainty because there is no analytical method to differentiate between biogenic and geogenic OC. We chose the range of 1,000-4,000 years because we concluded that this is a reasonable and a kind of worst case range of the mean 14C age of biogenic OC in the sediments. If we would use 10,000 yrs, this would mean that OC in the sediments would be derived only from biomass that was produced in the very beginning of pedogenesis 10,000 yr ago with no contribution of younger OC. This is unrealistic since sediments are no closed systems and will receive OC input if there is vegetation and pedogenesis (thus throughout the period of 10,000 years). Similarly, if we would choose the biogenic part of OC in the sediments to be only 100 yr old: DOC

measurements in the subsoil show 14C ages of several 1000 years. Surprising enough, the assumptions on the age of biogenic OC has little influence on our conclusions on the GOC fraction. Assuming that biogenic OC in the sediment has a mean of 10,000 yrs would not change a lot as can be seen in the following comparison. We took an average biogenioc OC age of **10,000 yrs** in the left Figure and a compared it to the mean age of **2,500 yrs** (middle of the 1,000-4,000 yr range) in the right Figure as we used in the manuscript:

[Figure]

Only the Miocene Sand with its low contribution from GOC would completely loose this geogenic fraction, while the magnitude for the Loess and the Red Sandstone only slightly decrease. We added this graphic to the supplement and a discussion about this uncertainty to 4.1:

4.1 "Generally, our calculations on the GOC fraction in the sediments are based on the assumption that biogenic OC in the sediments is not older than 4,000 yrs BP on average. And we also excluded the influence of a biogenic OC fraction that derives from soils that developed before the latest glacial period. Thus, there is uncertainty of a biogenic OC fraction in the sediments since it is unknown when biogenic OC entered the sediments. We assumed a mean age of 1000 to 4000 years based on DO14C data that was leached from the soil. Nevertheless, even with an assumed age of 10,000 years for the biogenic OC fraction, the highest possible contribution was 15 % for the Loss (94 cm depth) and 22 % for the Red Sandstone (74 cm depth) (Fig. A5). A mean age of 10.000 years would be an unrealistic assumption since sediments are open systems and may receive OC input throughout the pedogenic period if vegetation is present and not only at the start of pedogenesis."

**Reviewer 1**

The whole manuscript has intensively been checked for language and grammatic (also regarding all of your grammar/wording hints), so there are a lot of changes regarding these shortcomings in the new manuscript (see "marked manuscript").

l.18 this sentence is not necessary in the abstract and implicit from l 16.

Thank you for this hint. However, we think that this sentence is necessary because otherwise one could think that GOC (mentioned in l. 16) could be the geogenic **and** the biogenic part of OC that is

stored in sediments. This has already confused many readers of the manuscript during preparation so we decided to give a clear definition right from the start.

l. 30-31. These statements of the GOC contribution to SOC should be phrased depth explicit.

We added a range of possible contributions from GOC for this depth increment instead of a mean value:

"Its possible contribution to subsoil OC stocks (0.3-1.5 m depth) ranges from 1 to 26 %in soil developed in the Miocene Sand, from 16 to 21 % in the Loess soil and from 6 to 36 % at the Red Sandstone site."

l.32 this interpretation is speculative and should be removed. Subsoils without GOC show also depth trends and you did not evaluate the overall strength of GOC to these 14C depth across a wide enough range of soil types. In consequence, l. 33-34 need revision too. (Dot missing in l. 34)

Yes we completely agree with the reviewer and point out in l. 35. GOC "may partly explain the strong 14C increase in subsoil". Thus, we agree with the reviewer that there are also other reasons for this depth gradient. We revised the section to make clear that the main focus of our conclusion is, that it is possible to influence ages of OC in subsoils, but not solely explaining the strong increases in 14C ages. We would keep the conclusion in l. 33-34. but rewrite it, because it should become clear at which sites GOC could become important. We changed the sentence in 33:

"This is could be particularly important in young soils on terrestrial sediments with comparatively low amounts of OC, where GOC can considerably contribute to total OC stocks."

l.49. grammar/wording
l. 51 grammar/wording
l. 59. grammar/wording
l. 64 grammar/wording

- We completely revised the manuscript with the help of a professional language editor to get rid of grammar and wording mistakes

l.77-78. This sentence is only partly correct since human activities cause much more soil and sediment redistribution than any natural process. So I would limit your statement to natural deposition processes, and for those specifically again on which soils have developed.

- Yes, we agree with the reviewer in that human activities cause much more soil and sediment redistribution than any natural process. However, this sentence is about the original deposition of sediments and not about re-distribution by humans. Even if sediments are re-distributed by human processes and soils developed on this disturbed material, they still have been deposited > 50,000 yrs BP without major impact of humans.

l.81-82. This sentence is out of place and breaks the flow of the paragraph.

- We deleted this sentence

l. 84-85. This statement is in parts speculative. GOC is resistant under the conditions of deposition, but you cannot assume much from that under any other (in particular surface) conditions.

- Yes, thats true and we agree. Yet, we think that this is a valid assumption and in the following we concluded that it coud be degraded when the circumstances change (l. 85 ff) due to the input of fresh water, air and microorganisms from above for example.

l.88 grammar/wording

l.91-93. Give this sentence a direction. Are you referring here to priming effects?

- Has been changed accordingly:

"If GOC is degradable in OC-poor sediments or sedimentary rocks has not been investigated so far but might be different since the amount of microbial biomass mediated by the OC content can also drive microbial respiration (Colman and Schimel, 2013). Therefore, these sediments might have less microorganisms, that could also be spatially separated from the GOC, which might hamper its respiration."

l.95 grammar/wording

l. 101-104. Question 1) and 4) seem related/repetitive. Combine

- Thank you for this comment. We guess, the questions were not formulated clear enough. Question 1 refers to the amount of OC in subsoils in comparison to the amout of total OC in sediments. Question 4 refers to the contribution of geogenic OC to subsoil OC. We made this more clear and now write:

- Question 1 refers to the amount of OC in subsoils in comparison to the amout of total OC in sediments. Question 4 refers to the contribution of geogenic OC to subsoil OC. We made this more clear.

"Our main research questions were i) what is the relation between the amount of  OC in the soil and in the sdiment? ii) is OC in sediments 14C free and how much is really geogenic? iii) will sedimentary GOC be degraded? and iv) how much does GOC from the sediments contribute to soil OC?"

l. 145 grammar/wording

l. 160 name again what acid at what concentration

- Added "1 % HCl "

l. 177 this "transition to sediments" does not match to what you show in your profile pictures as beginning C horizons.

The transition between soil and sediment in the profiles was gradual and we detected pedogenic influence also below 80 cm depth as indicated by the "v" for the C horizon. The visible „Cv" horizon

represents the transition between soil and sediments, but not the start oft the sediment.

l.178 grammar/wording

l. 214-215. Revise the added sentence for clarification. Maybe add "…degradation rates at lower temperatures".

- Was changed accordingly.

l. 213. What was the incentive to incubate the crushed sandstone for 63 days, but the non-crushed for 50 days?

- There was no intention to incubate for different time periods. It was just due to methological circumstances.

l. 225 unclear what you mean with "poured bulk density".

- We added the explanation:

"A water content corresponding to 40 % of the water holding capacity based on the poured bulk density, determined by filling the loose material into a defined volume and measuring its weight, was adjusted"

l. 293-294. This strange finding is interesting, but seems to be an outlier when looking at figure 2? Has this measurement been confirmed? Or checked with measurements of a similar depth for this sample? It seems to drive your OC stock calculation in table 1 and needs to be confirmed and checked very carefully.

Unforntunately we are not sure which data point(s) you refer to. Every measurement of OC in the sediment was based on samples from the same depth but different cores to check for variability. Unfortunately, we only had one Loess core so the depth variability could not be confirmed with two cores in this case. However, the measurement in 4 m depth could be confirmed visually. The sample from this depth showed a very dark colour, pointing out to its high OC content. We added a sentence in 3.1:

"This was no outlier because the high OC content could be confirmed visually by the very dark colour of the sample"

We further added a sentence in 4.1.1:

"For example, the very dark sample in 4 m depth with its high OC content points out to sedimentation circumstances that favoured the accumulation and preservation of OC."

l. 287-311. Figure numbering refers to the former version.

- Will be changed. Thanks for noticing.

l. 302 grammar/wording

l.333-334. This statement is discussion and interpretation, not results.

- Deleted from l. 333 and added following sentence to 4.1.1:

"Furthermore, the modern like 14C signature in 21 cm depth could be due to the plough layer at the Loess site mixing up the upper 30 cm."

l.339 grammar/wording

l. 326-353. Also in this section the figure numbering seems to refer to the old MS version. Please check throughout the MS that his is addressed. The same is the case in the next section, I stopped checking after that one.

- Thank your. All figure numbers have been checked.

l. 375 grammar/wording

l. 378-380. This sentence is important but seems to be out of place here. Better placed in the methods? Besides, you state in your responses that O2 was >20% throughout the incubation experiment. How have you checked that ? Related to this Figure A4 panels are not well described. To me it does look like you have CH4 production increasing over time. In this regard, I would then make sure to not use the 553 days datapoint in the manuscript (Figure A2) for any interpretation or calculations (it is also not constrained like the other points shown in Figure A3).

- Thank you for this suggestion. We agree with the reviewer that this sentence would also fit into the materials and methods section. However, we decided to keep it in the result section since we are referring to the measurement results of the incubation. In l. 378-380 we stated that CH4 development remained on a low level, indicating no oxygen limitations.

Concerning your question about $O_2$ concentrations: We did not measured $O_2$, but we had a large headspace in the incubation vessels varying between 4,200 and 4,900 $cm^3$ and additionally a rather low microbial activity in the vessels (very low mineralisation rates). Furthermore we started with 20 % O2 and measured CO2, N2O and CH4. Therefore we assume that there was no oxygen limitation. So it was no direct measurement but an indirect conclusion. At least there was a slight slight increase in CH4 over time but only for the Red Sandstone samples. The ration between produced CH4 and CO2 was however always below 0.03 indicating that the incubation was mainly aerobic.

We will remove Fig. A4 from the supplement, because it does not contain any further information.

l. 469. A reason why an annotated photo figure of the cores would be important (see comment at "figures").

- Yes, we agree that a photo would be nice. However, since the Loess and Miocene Sand cores were obtained in closed tubes we could not make a photo of the core material. We could just see the

material when we selected samples from the inside. We added a Fig. A6 with pictures from the Red Sandstone samples in the supplement.

l.470-471. grammar/wording

l.474-478. This statement needs a proper reference. DOC infiltration rates have not been measured in any of the substrates.

Thank you for this valuable comment, we canged the sentence accordingly to:

"A loosely bedded sediment like the Miocene Sand with extremely low concentrations of OC could be more prone for infiltration of biogenic OC and dilution of GOC."

Because we want to draw the attention to the low OC contents in the sediments and not so much towards infiltration rates in different sediments.

l. 515-549. What are typical subsoil temperatures here? MAT? And its not only the temperature. Its also aeration, water content, structure etc. So I don't think your experiment can reveal too much on the decomposability of deep C in situ but you can compare to other similar experiments or what the mineralization rates would be under conditions similar to your experiments. Revise section accordingly and reduce to avoid speculation.

- We added a reference for typical subsoil temperatures in 4.2

"For subsoils with comparable climatic conditions, Wordell Dietrich et. al (2020) found seasonal temperatures in 150 cm depth ranging from 4 to 14.4°C over a 2 year period."

 In l. 212 ff (M&M) we already described the experimental design and mentioned that we want to assess the „potential stability of OC" and not the stability under in situ conditions. What we did was to compare the incubation experiment with other experiments under similar conditions in l. 526 ff.

l. 586. Remove "long-term" since you work without the uncertain 533 days datapoint, no?

- Was removed accordingly

l. 612-613. grammar/wording

l. 631-632. This 30% statement is speculative and unnecessary. Remove sentence.

- Was removed accordingly

l. 653 grammar/wording

l. 655 grammar/wording

l. 677. Delete "seem to"

- Was removed accordingly

l. 678. Add a sentence that your experiment shows that the GOC is degradable under the conditions you created during your incubation, no?

General comment discussion: The discussion is quite long. That's OK as it is mostly data focused. However, I am missing a bit of a critical view on the assumptions behind the 14C signatures used to calculated the biogenic and geogenic. Uncertainties are discussed here and there a bit but there are some really big assumptions here with DOC and root exudates being involved over assumptions on what age range to expect from biogenic OC and so on. A critical view on this before going into all the data interpretation details would be helpful for readers to get a better feeling on the uncertainties surrounding the data.

 - Thank you for this important note. We added a section at the beginning oft he discussion regarding the unertainty of our underlying assumptions with 14C (at 4.1). Also see: response to editor comment 2)

General comment conclusions: I like the conclusion a lot. It really focuses on what the data can clearly show. It makes me think however, why all the data from the very deep cores was discussed in such great detail in the first place. Frankly, the manuscript would profit on more focus on the soil part, and not so much of the rather (speculative) mineralization or origin discussion for the deeper sediment cores (I see this as rather supplemental).

- Thats true. The incubation experiment has a long part in the discussion. We will shorten the discussion for these parts.

Comments on figures and tables:

Figure 1. Reformat Figure. Replace scale bars and signs with a digital version. Write horizon labels either on the left or to the right but not central into the image. Cut image to focus area you want to display. Use USDA or WRB system of horizon classification, not KA5. Besides, I cannot follow the classification easily and also some of the interpretation later in the text when relating to it. Some examples: Al horizon features at 66cm, or even 82cm depth (image A). How is that possible? If there was no disturbance in the forests, where is the rather big M horizon coming from (image B) which is also not discussed anywhere. Why would the sandstone profile (image C) have the most shallow development depth? Furthermore, since your MS is about long sediment cores, please add annotated images of the cores too (in the supplement), especially as you stress that the sediments were homogenous for 10m.

- Thank you for this detailed analysis of the soil profiles and your suggestions for reformatting. In general Fig. 1 was added to give a short overview about the developed soil profiles. We think that a complete reformating of Fig. 1 as suggested would be without substantial benefit since we are not discussing the soil profiles in detail. The M horizon from 1B propably derives from erosion of soil material down the slope. Unfortunately, we can not add images of the loess and the sand cores since

they were obtained in closed cylinders and we only removed material from the inside. We will add pictures from the Red Sandstone cores in the supplement. We furthermore described the classification of the soil in more detail:

"The soil is classified as a Folic Brunic Arenosol according to the World Reference Base for Soil Resources (WRB, 2006). The sediments were loessic deposits (Weichselian Glacial) that have been under agricultural landuse for the past decades, 30 km north of Göttingen (51°48.101 N; 9°58.002' O), referred to as "Loess",and terrestrial sandy deposits from the Miocene (Neogene formerly named Tertiary) in a European beech forest 13 km south-west of Göttingen (51°28.673 N; 9°45.323' O) referred to as "Miocene Sand". The respective soils have been classified as a Haplic Luvisol and a Dystric Chromic Arenosol accordingly. "

Figure 2. Some of the panels have weir line features at the x axis and y axis zero lines. Remove them. Further, red sandstones cores seem to be quite different from each other. This contradicts to your statements of the homogeneity of the core samples. How is this addressed?

- Changed Fig. 2 accordingly. The inhomogenity of the Red Sandstone samples was adressed in l. 305 („Unexpectedly high amounts of inorganic carbon IC were found in parts of the Red Sandstone, indicating the presence of calcareous deposits in this terrestrial material (Fig. 1 c)"). The statement of homogenity does not mean that the cores did not show any variation with depth. There were calcareous layers in the Red Sandstone and the Loess was very unhomogeneous overall. But it was more or less the same material down to 10 m depth. We thank the reviewer for addressing this important point and now describe this incomplete homogenity in more detail in the result section in 3.1:

 "Despite having the same material down to 10 m depth at each site, there were still some inhomogeneities visible and also measurable. This was especially true for the Loess and the Red Sandstone."

Figure 5 essentially shows the same as Figure A3. I think you can remove Fig. A3

- You probably mean Fig. A4 (A3 is methane production during the incubation). A4 was removed.

Table 2: Make sure the mineralization rates using the linear model are done based on the 63 day timeframe of the experiment and not include the uncertain extrapolation to the 533 days. Unclear from Fig A2.

- In Fig. A2 no interpolation model was used as it is mentioned in the figure caption. It should not be confused with the 63 day linear interpolation used for Fig. 4.

**Review 2**

Major points to be addressed:

- Is there something known about the origin of GOC in the three different sites (i.e.

buried organic matter / soot / coal / …)? As it is not stated/hypothesized, it makes also

difficult to evaluate the used temperatures (450 ℃) for preheating samples to remove

OC.

- We do not have detailed information about the origin of GOC in the sediments. Due to their d13C values we concluded that they have a plant origin. And additional measurements (e.g. NMR) revealed inaformation about the chemical composition of O . We added information about the deposition in the site description:

"The sediments at the Loess site were deposited between the last glacial and interglacial periods between 115,000 and 400,00 yrs BP according to Jordan and Schwartau (1993)."

Additionally we added a reference to evaluate the used temperature of 450°C.

- Line 54: "GOC in most cases is devoid of 14C and thus may lead to an overestimation

of ancient OC sources although a number of studies showed the importance of root

derived, young OC inputs to subsoils." This is quite a fundamental point of your study

and could be better stated here already. Suggested: "As 14C has a half lifetime of 5730

years, carbon deposited from the Weichselian and older are depleted in 14C, thereby

diluting the overall 14C concentration. Especially in C poor subsoil, where GOC

forms a relative larger part of the overall C content, this leads to an age overestimation

of relative fresh OM, like root derived components. "

- We added a sentence to make the influence of GOC on 14C concentrations in subsoils more clear: "Therefore especially in OC poor subsoils, GOC may significantly influence and dilute the overall 14C signal."

Thanks for the suggested formulation! But the sentence *„Especially in C poor subsoil, where GOC forms a relative larger part…"* is already a conclusion which we would prefer not to make at this point. Furthermore if GOC forms a relative large part of GOC this would not lead to an *„age overestimation"* of fresh OM but dilute the overall signature of OC which is a difference.

- Line 80: "Thus, using both carbon isotopes can reveal if the OC is a mixture of GOC

and OC". It is not clear how δ13C can be used (from the introduction) and more detail

how these different isotopes can be used to disentangle the different C components

should be added. Above this line it is only made clear why GOC and 14C are

important to study.

- We changed the sentence in l. 80 to make it more clear how the d13C signature can be used:

"In addition, $\delta^{13}$C values of OC in the sediments allow to distinguish carbonaceous with its $\delta^{13}$C values around 0 ‰ from organic sources with $\delta^{13}$C values < -22 ‰"

- Line 101-102: Restructure and rephrase questions, especially as question II is

fundamental for the  disentanglement of geogenic and more recent OC. It is suggested

to start with "Is (G)OC free of 14C", than "how much does GOC contribute to

(sub)soil OC?" and "will sedimentary GOC be degraded and/or incorporated in recent

OC"

Changing Question 1 to „Is (G)OC free of 14C" would not be valid, since the geogenic part of OC is by definition free of 14C because it was sedimented more than 50.000 years ago. We would prefer to keep the structure of the questions since the structure of our whole manuscript is based on these questions.

- Line 261-266: This part of the results does not create confidence in your data. First it

is stated all samples were within detection limit (to my opinion an understatement, as

otherwise samples should not be included or represented by the value 0) and next there

is speaking of "random noise". Better to simply state what the mean relative standard

deviation was (or overall measurement/methodological error) and the lowest measured

value (0.04 g C kg-1 soil). Note that mg C g-1 soil and g C kg-1 soil are both used in

the text.

 - We changed the first sentence accordinglyto make it more clear what we mean (there were no samples that contained no OC). The „random noise" expression will be changed. We will change all „mg C g-1 soil" to „g C kg-1 soil", thanks for noticing!

"In all analysed sediments measurable OC contents were detected"

"Thus the range from 0.00 to 0.01 mg C $g^{-1}$ soil was assumed to be mean standard error from the measurement"

- You could consider to discuss first the "How much GOC contributes to soil organic

carbon?" before going into the bioavailability of it. This would make the "flow" of the

discussion more logical.

- Yes, we agree with the reviewer that this would also be possible. Still we think it makes more sence to put the degradation part first. Thats because if we want to conclude how much GOC contributes to subsoil OC we have to know how much could have been mineralised. If we would write it the other way around, we would write about the GOC contribution 2 times. First under the assumpion that there is no degradation and then about the contribution when there is degradation.

Minor suggestions for improvement:

Line 22: "this gap" -> this knowledge gap

- Changed accordingly

Line 24: "sedimentary OC" -> GOC

- Thanks for the suggestion, but in this context this would be wrong since we are referring to the bulk OC in the sediment and not just the geogenic part.

Line 51: "an contribution" -> "a contribution"

- Changed accordingly

Line 60: "have been investigated" Missing the results of these studies, probably rephrase.

- Changed sentence to:

"Furthermore OC rich sediments with contents of 2-7 g kg$^{-1}$ (Hemingway et al., 2018) or 28-105 g kg$^{-1}$ (Frouz et al., 2011) have been investigated with regard to the stability of OC in these sediments but with no conclusion for GOC contributions in soils."

Line 68-69: "more information about the amounts of OC in sediments is needed." -> "GOC in

sediments" or "contribution of GOC in sediments"

- Changed accordingly

Line 74: "hydraulic conductivity" -> "Pore distribution" or "porosity" fits the context better.

- Changed accordingly

Line 127-128: "This means e.g. for a sample increment from 1-2 m, the sample represents the

1.85-1.95 m depth" -> "This means that for example the increment 1-2m is represented by a

sample from 1.85-1.95m depth."

- Changed accordingly

Line 148: "removing carbonates" Same as 14C?

- We deleted the sentence here because sample treatment for 14C is described in 2.3 more detailed

Line 250: "lm" -> "the function lm"

- Changed accordinly

Title 3.1: "…sedimentary and subsoil organic carbon" -> be consistent with terminology.

Better to use GOC / geogenic organic C instead of "sedimentary", especially as "subsoil

organic carbon" can be all OC found in the subsoil

- Yes, subsoil organic carbon be all OC found in the subsoil. But that is what we mean in this chapter. We are referring to the comparison between the bulk OC content in the subsoils and the bulk OC content in the sediments with no distinction between the geogenic and the biogenic part.

Line 296/277: Fig 1a. -> Fig 2a.

- Changed accordingly

Line 284 "they all were in the range of C3 plant material. A value above -25 ‰ for the Red

Sandstone in 4 m depth can be explained by corresponding high values of inorganic carbon

(IC) in this depth" -> better for discussion

- Yes that's true. But we are not discussing the 13C signatures in the discussion section and only use it for the distinction of biogenic and calcareous C. Because the discussion is already quit long we would prefer to keep this sentence in the result section.

Line 308: "Fig. 2 a" -> "Fig. 3a"

- Changed accordingly

Line 322: "Fig. 2 c" -> "Fig. 3c"

- Changed accordingly

Line 424: "the same site assigned the different" -> "the same site and assigned the different"

- Changed accordingly

Line 425: "sedimentary OC" -> "OC"

- Leaving out "sedimentary" in this context could be misleading, since we are only referring to the sediment and not to OC in the soil.

Line 426/427: "…extremely low concentrations of OC is more prone for infiltration of

biogenic OC" Not completely clear what is meant, but probably best to say:"… very low OC

contents increases the relative importance of biogenic C input for the over OC"

- Sentence has been changed accordingly to make it more clear:
"A loosely bedded sediment like the Miocene Sand with extremely low concentrations of OC could be more prone for infiltration of biogenic OC and dilution of GOC."

Line 463: "sedimentary bedrock" Loess is no bedrock, but an (aeolian) deposit or sediment

- changed to "sediment"

Line 523: Not clear what is meant with "a resistant part"

- Changed to:

"Hemingway et al. (2018) found that sedimentary OC directly exposed to the surface in a rapidly eroding tropical mountain area exhibits a considerable mineralisation down to 1 m below the surface also leading to around 30 % of GOC remaining in the soil."

Line 525: "distinguished" -> "Distinguish"

- Changed accordingly

Line 526: "bedrock OC" -> "GOC"

- In this context we are again referring to the bulk OC in the sediment and not just to the geogenic part.

Line 557: "despite differences between sediments" -> ", despite differing between sediments,"

- Changed accordingly

Line 570-571: Combine sentences

- Changed accordingly

Line 572: "high age" -> "high 14C age"

- Changed accordingly

---

## Author Response (AR3)

**Revision reply**

**Editor comment**

Your manuscript may be acceptable now for publication. However, your abstract still needs improvement:

1.) 'However, the contribution of GOC to total soil OC varies depending on the type of bedrock. As yet, no far studies have investigated the contribution of GOC derived from different terrestrial sedimentary rocks to soil OC content' should be changed to 'The contribution of GOC to total soil OC may vary depending on the type of bedrock. However, no studies have been carried out to investigate.....'

- Was changed accordingly

2.) Please add a more detailed description of what was done to the abstract - this refers in particular to the analyses of the 1m depth intervals and your calculation of the geogenic and biogenic components in particular, the references used to calculate both. What is your definition of recent carbon?

- We added more detailed information about the analysed depth intervals and the calculation of biogenic and geogenic components to the abstract. We changed the sentence in l. 23 (Marked-up manuscript) to:

"In order to fill this knowledge gap, 10-m long sediment cores from three sites recovered from Pleistocene Loess, Miocene Sand and Triassic Red Sandstone were analysed in 1 m depth intervals and the amount of GOC calculated based on $^{14}$C measurements"

We further changed the sentence in l. 26:

"The biogenic component relates to OC that entered the sediments from plant sources since soil development started."

And added a detailed description in l. 28:

"Assuming an average age for this biogenic component ranging from 1,000-4,000 years BP we calculated average amounts of GOC in the sediments starting at 1.5 m depth based on measured $^{14}$C ages. The median amount of GOC in the sediments was then taken and its proportion of soil mass (g GOC per kg$^{-1}$ fine soil) calculated in the soil profile."

3.) In addition, in the concludion a short description of your approach including its underlying hypotheses should be stated. 'This approach' is not enough as it is not understandable what 'this' refers to.

- We added two sentences at the beginning of the conclusion to define the aim of ouf study and describe our approach:

"In this study the amount of GOC in sediments and in the soil was analysed by radiocarbon dating. The aim was to find out if GOC from different terrestrial sediments can have an influence on soil OC stocks."

39    4.) 'Subsoil' should also be redefined here shortly.

40    - We added the definition for subsoils defined in our study in l. 618:
41    "These amounts allowed for contributions from GOC of between 10-30 % in subsoils, defined here as
42    soil horizons ranging from 0.3 to 1.5 m depth."

43    5.) 'Thus, even sediments with comparatively low amounts of OC were also able to demonstrate the
44    large contribution of GOC'. This sentence needs to be reformulated or deleted - it is not clear what is
45    means and why it is important here.

46    - Sentence was deleted